# Photobactericidal activity activated by thiolated gold nanoclusters at low flux levels of white light

Gi Byoung Hwang[1], He Huang[2], Gaowei Wu[2], Juhun Shin[1], Andreas Kafizas[3,4], Kersti Karu[1], Hendrik Du Toit[2], Abdullah M. Alotaibi[1], Layla Mohammad-Hadi[5], Elaine Allan[6], Alexander J. MacRobert[5], Asterios Gavriilidis[2] & Ivan P. Parkin[1✉]

The emergence of antibiotic resistant bacteria is a major threat to the practice of modern medicine. Photobactericidal agents have obtained significant attention as promising candidates to kill bacteria, and they have been extensively studied. However, to obtain photobactericidal activity, an intense white light source or UV-activation is usually required. Here we report a photobactericidal polymer containing crystal violet (CV) and thiolated gold nanocluster ($[Au_{25}(Cys)_{18}]$) activated at a low flux levels of white light. It was shown that the polymer encapsulated with CV do not have photobactericidal activity under white light illumination of an average 312 lux. However, encapsulation of $[Au_{25}(Cys)_{18}]$ and CV into the polymer activates potent photobactericidal activity. The study of the photobactericidal mechanism shows that additional encapsulation of $[Au_{25}(Cys)_{18}]$ into the CV treated polymer promotes redox reactions through generation of alternative electron transfer pathways, while it reduces photochemical reaction type-II pathways resulting in promotion of hydrogen peroxide ($H_2O_2$) production.

[1] Materials Chemistry Research Centre, Department of Chemistry, University College London, 20 Gordon Street, London WC1H 0AJ, UK. [2] Department of Chemical Engineering, University College London, Torrington Place, London WC1E 7JE, UK. [3] Department of Chemistry, Imperial College London, Molecular Science Research Hub, White City Campus, 80 Wood Lane, London W12 OBZ, UK. [4] Grantham Institute, Imperial College London, Exhibition Road, London SW7 2AZ, UK. [5] UCL Division of Surgery and Interventional Science, Royal Free Campus, Rowland Hill Street, London NW3 2PF, UK. [6] Department of Microbial Diseases, UCL Eastman Dental Institute, University College London, 256 Grays Inn Road, London WC1X 8LD, UK. ✉email: i.p.parkin@ucl.ac.uk

The discovery of antibiotics in the early 20th Century revolutionised the treatment of bacterial infections but after several decades of overuse, the evolution of antibiotic resistant bacteria is a major threat to the practice of modern medicine[1–7]. Surfaces in healthcare facilities used to be considered not to have an effect on the spread of infection[8]. Recent studies have reported that bacteria-contaminated surfaces make a significant contribution to the incidence of healthcare associated infections[8–13], and that repeated cleaning/disinfection of the contaminated surfaces, which is commonly performed in healthcare facilities, is not always sufficient to remove pathogens from surfaces. For example, 27% of surfaces in rooms were still contaminated by bacteria after four complete cleaning cycles with disinfectant[14–16].

Photobactericidal agents have gained significant attention as promising candidates to kill bacteria and keep surfaces sterile. Toluidine blue O (TBO), methylene blue (MB) and crystal violet (CV) are widely used for surgical and biological stains and wound disinfection, and are known to have photobactericidal activity[17]. When the agents are exposed to a bright visible light source, they are photoexcited and induce reactive oxygen species (ROS), killing bacteria through damage to the cell membrane and DNA[18–22]. Due to their photobactericidal property, and ease of use, the photobactericidal agents were used to treat infections of blood products and oral infections, and are also effective at killing antibiotic resistant bacteria[22].

It has been shown that zinc oxide (ZnO), silver (Ag), and gold (Au) nanoparticles (NPs) in combination with TBO, CV, and MB produce a synergistic bactericidal effect. This effect was most pronounced using intense light sources such as lasers or bright light of white light source (1000–10000 lux; 0.15–1.5 mW cm$^{-2}$)[23–28]. For gold NPs, there is a size effect, where the smallest NPs tested (2–20 nm) showed the greatest promotion of the dyes' bactericidal properties[23–29]. The 2-nm Au NPs are at the crossover point between the quantum description of matter and a molecular cluster approach[30].

Gold clusters consist of a small number of atoms, and its typical size is less than 2 nm[31,32]. A Au cluster has discrete physico-chemical properties compared to larger Au NPs. In contrast to Au NPs, the cluster does not show surface plasmon resonance, and the electronic band structure of the cluster is different from Au NPs, where the band structure of Au NPs is typically continuous whereas Au nanoclusters form a discontinuous band structure with discrete energy levels[33].

The motivation herein was to see if discrete molecular gold clusters could promote a greater synergistic effect in enhancing the photobactericidal effect of CV dye. To evaluate this, we synthesize $[Au_{25}(Cys)_{18}]$ clusters via a scalable microfluidic approach and incorporated them with CV into a polymer surface to test its bactericidal properties. Surprisingly, this cluster promotes bactericidal activity at very low levels of white light. We suggest that the presence of $[Au_{25}(Cys)_{18}]$ promotes redox reactions by generating an electron transfer pathway from CV to the cluster, resulting in enhanced $H_2O_2$ formation and bactericidal activity.

## Results and discussions

To continuously synthesize $[Au_{25}(Cys)_{18}]$, a microfluidic segmented flow system that consisted of a tube-in-tube membrane contactor system and coil flow inverter reactor was used (Supplementary Fig. 1). In this system, Au (I) solution containing 3 mM cysteine was mixed with CO-saturated heptane, which was prepared by the tube-in-tube; The mixture passed through a coil flow inverter reactor at 80 °C. In the reactor inverter, $[Au_{25}(Cys)_{18}]$ was continuously synthesized with residence time

of 3 min. The $[Au_{25}(Cys)_{18}]$ solution was stored at room temperature overnight. After that, an organic layer, formed by heptane, on the top of the solution was removed. As shown below, this process exclusively formed $[Au_{25}(Cys)_{18}]$ clusters with no evidence of larger or smaller clusters.

The UV–Vis spectrum of the $[Au_{25}(Cys)_{18}]$ solution was measured from 300 to 900 nm. As shown in Fig. 1a, the nanoclusters gave multiple absorption peaks at 400, 450, 670 and 780 nm which are in agreement with the absorbance features of thiolated $Au_{25}$ clusters reported previously[34]. It was confirmed that the concentration of synthesized $[Au_{25}(Cys)_{18}]$ was about 0.12 mM ($7.2 \times 10^{16}$ $[Au_{25}(Cys)_{18}]$ clusters mL$^{-1}$). ESI-MS was used to determine the molecular composition of $[Au_{25}(Cys)_{18}]$. As shown in Fig. 1b, it was observed that the most intense set of peaks were at $m/z$ ~2361.4 over a range of $m/z$ 1500–5200, and Fig. 1c shows that the base peak at $m/z$ 2361.4 (peak #1) was accompanied with a group of similar small peaks (peak #2 to #7) corresponding to $H^+$ dissociation and $Na^+$ coordination to $[Au_{25}(Cys)_{18}]$. Isotope pattern analysis of peak #1 showed that the peak spacing between $^{12}C$ and $^{13}C$ was about 0.33 indicating that $[Au_{25}(Cys)_{18}]$ carried three negative charges resulting in the generation of $[Au_{25}(Cys)_{18}-3H]^{3-}$ with molecular weight (MW) of 7083.2 Da (Fig. 1d). Other ionised species (#2 to #7) are (#2) $[Au_{25}(Cys)_{18}-4H+Na]^{3-}$ (MW 7106.1), (#3) $[Au_{25}(Cys)_{18}-5H+2Na]^{3-}$ (MW 7128.1 Da), (#4) $[Au_{25}(Cys)_{18}-6H+3Na]^{3-}$ (MW 7150.2 Da), (#5) $[Au_{25}(Cys)_{18}-7H+4Na]^{3-}$ (MW 7172.2 Da), (#6) $[Au_{25}(Cys)_{18}-8H+5Na]^{3-}$ (MW 7194.1 Da) and (#7) $[Au_{25}(Cys)_{18}-9H+6Na]^{3-}$ (MW 7216.2 Da). The ionised species and isotope patterns containing #1 to #7 were identical to those obtained by theoretical simulation (Fig. 1d and Supplementary Fig. 2). Additionally, other sets of cluster peaks were observed at $m/z$ ~1808.6, 2411, and 3542.6, indicating 2-, 3- and 4- charged gas phase ions. All peaks corresponded to $[Au_{25}(Cys)_{18}]$, indicating that no other clusters were formed[35]. TEM (Fig. 1a inset) shows that the size of the $[Au_{25}(Cys)_{18}]$ clusters was <2 nm[36].

A swell-encapsulation-shrink process was employed to produce photobactericidal silicone. Fluorescence microscopy, UV–Vis spectroscopy and XPS confirmed that after the treatment, CV molecules and $[Au_{25}(Cys)_{18}]$ clusters penetrated into the polymer matrix (Supplementary Fig. 3). As shown in Fig. 2a, after $[Au_{25}(Cys)_{18}]$ or cysteine treatment, the silicone substrate maintained its colour, while CV treatment changed from white to violet. The colour of the polymer containing CV&$[Au_{25}(Cys)_{18}]$ was more intense than the sample with CV alone. Figure 2b shows UV–vis absorbance spectra of the control and treated samples at a wavelength of 400–800 nm. All of the CV-treated polymers have a main absorbance at 595 nm with a shoulder peak at 541 nm. The absorbance of $[Au_{25}(Cys)_{18}]$ encapsulated sample was broader and of higher intensity compared to the sample with CV alone. Of materials used for the encapsulation, CV is the only material containing chlorine (Cl). Thus, a change in the weight percentage (wt%) of Cl was determined before and after additional encapsulation of $[Au_{25}(Cys)_{18}]$ by X-ray fluorescence spectroscopy (XRF). Cl wt% in CV&$[Au_{25}(Cys_{18})]$ polymer was 1.8 times higher than the polymer containing CV alone. This was similar with the increase of light absorbance of the material at 595 nm after additional encapsulation of $[Au_{25}(Cys)_{18}]$ into the CV-treated polymer. This showed that addition of $[Au_{25}(Cys)_{18}]$ enhanced CV impregnation into the polymer. The image analysis of sliced polymers supports the increase of CV uptake by $[Au_{25}(Cys_{18})]$ (Supplementary Fig. 4). We speculate that this is because negatively charged $[Au_{25}(Cys)_{18}]$ (zeta potential of $[Au_{25}(Cys)_{18}]$ solution: −31.8 mV) attracts more $CV^+$ during the encapsulation process[37].

*Staphylococcus aureus* and *Escherichia coli* are common pathogens in healthcare facilities and can cause a range of infections from local skin and catheter-related infections to sepsis

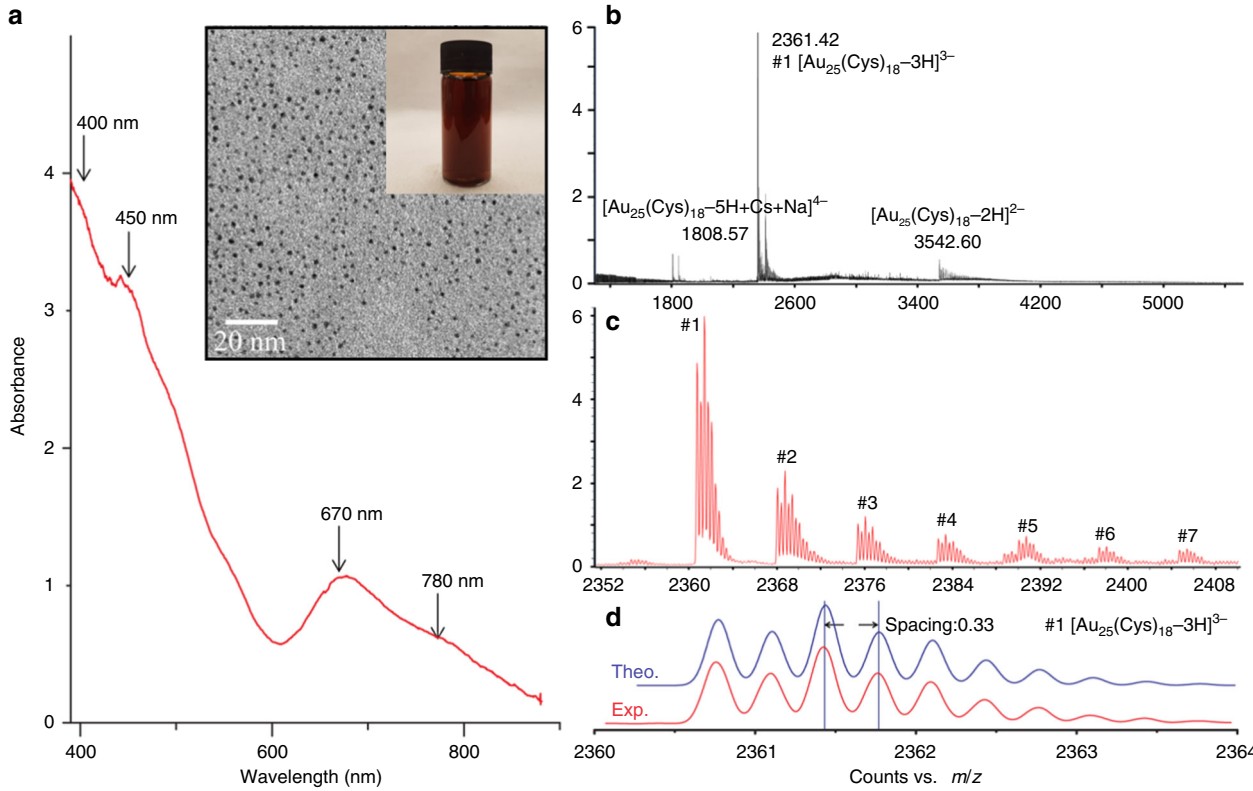

**Fig. 1 Characterization of synthesized [Au$_{25}$(Cys)$_{18}$]. a** UV–Vis spectrum of [Au$_{25}$(Cys)$_{18}$] solution (inset shows the picture and TEM image of [Au$_{25}$(Cys)$_{18}$] solution). **b**, **c** ESI-mass spectra of [Au$_{25}$(Cys)$_{18}$]. **d** Comparison between theoretical and experimental isotope pattern of #1 peak.

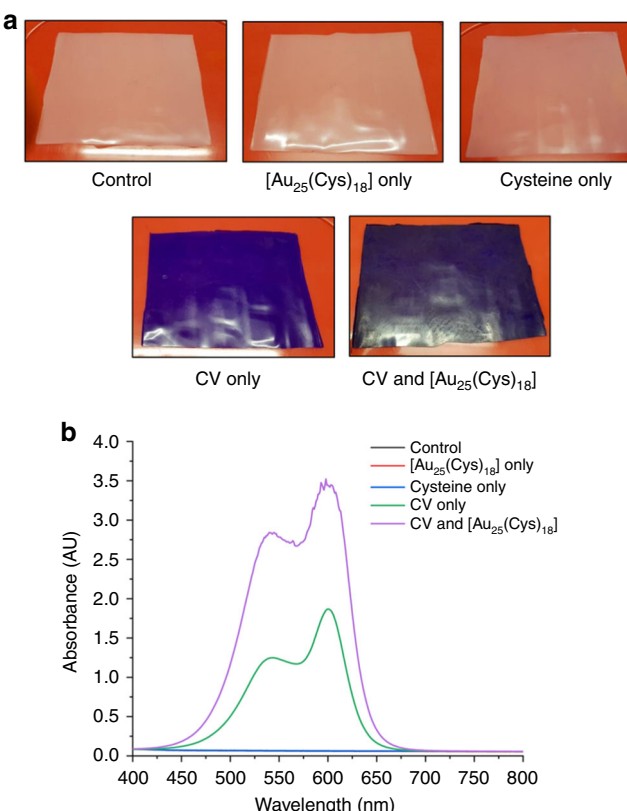

**Fig. 2 Images and UV–Vis measurement of 24 h treated silicones. a** Images of control, CV only, [Au$_{25}$(Cys)$_{18}$] only, cysteine only and CV& [Au$_{25}$(Cys)$_{18}$]. CV indicates crystal violet. **b** UV–Vis absorbance spectra of control, CV only, [Au$_{25}$(Cys)$_{18}$] only, cysteine only and CV&[Au$_{25}$(Cys)$_{18}$].

and even death of patients[38,39]. The bactericidal activity of the polymers against *S. aureus* 8325-4 *and E. coli* ATCC 25922 was tested in the dark and under white light (average 312 lux), and the result was compared to that of a control comprised of solvent treated silicone. Twenty-five microlitres of the bacterial suspension containing ~$10^5$ CFU mL$^{-1}$ was inoculated on to each sample surface and then exposed to the white light source or maintained in a dark room at 20 °C. Supplementary Fig. 5 shows the distribution of white light intensity. The intensity ranged from 200 to 429 lux (0.03 to 0.06 mW cm$^{-2}$) and the light emission wavelength mainly ranged from 400 to 800 nm[40].

Figure 3a, b shows the bactericidal activity of the samples against *S. aureus* after 6 h incubation in the dark and in white light. After 6 h incubation in the dark, compared to the control, a reduction in the number of viable bacteria was not observed on either silicone samples containing only CV or [Au$_{25}$(Cys)$_{18}$] alone, whereas a statistically significant reduction in bacterial numbers was observed on the silicone samples containing cysteine only (0.17 log reduction, *T*-test: *P* < 0.05) and CV& [Au$_{25}$(Cys)$_{18}$] (0.25 log reduction, *T*-test: *P* < 0.1). Upon 6 h exposure to white light, a reduction in the number of viable bacteria was not observed on samples containing CV only or [Au$_{25}$(Cys)$_{18}$] only, compared to the control. Bactericidal activity of silicone containing cysteine alone was observed after 6 h exposure to white light, but the activity was not enhanced compared to that in the dark (*T*-test: *P* > 0.1). The bactericidal activity of the combined CV&[Au$_{25}$(Cys)$_{18}$] encapsulated sample was significantly enhanced after 6 h exposure of white light; the number of viable bacteria on the sample with CV&[Au$_{25}$(Cys)$_{18}$] fell to below the detection (<$10^2$ CFU mL$^{-1}$), indicating a >3.3-log reduction in viable bacteria.

Figure 3c, d shows the bactericidal activity of the samples against *E. coli* after 24 h incubation in the dark and in white light. After 24 h incubation in the dark, compared to the control, the

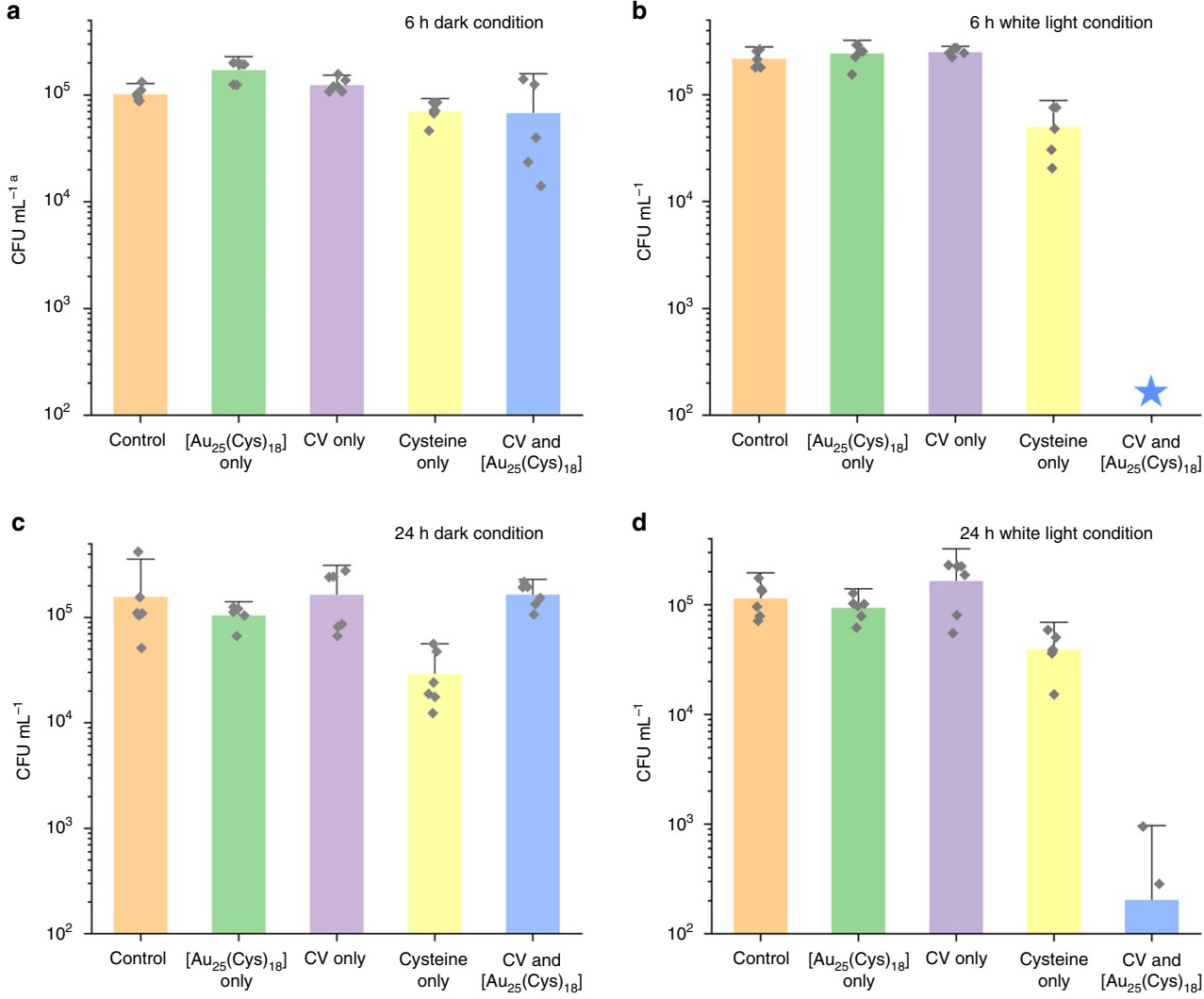

**Fig. 3 Bactericidal activity of control and treated silicone samples. a** Bactericidal activity of control, [Au$_{25}$(Cys)$_{18}$] only, CV only, Cysteine only and CV&[Au$_{25}$(Cys)$_{18}$] against *S. aureus* in dark (*n* = 6 independent samples). CV indicates crystal violet. **b**, Bactericidal activity of control, [Au$_{25}$(Cys)$_{18}$] only, CV only, Cysteine only and CV&[Au$_{25}$(Cys)$_{18}$] against *S. aureus* in white light (*n* = 6 independent samples). **c** Bactericidal activity of control, [Au$_{25}$(Cys)$_{18}$] only, CV only, Cysteine only and CV&[Au$_{25}$(Cys)$_{18}$] against *E. coli* in the dark (*n* =6 independent samples). **d** Bactericidal activity of control, [Au$_{25}$(Cys)$_{18}$] only, CV only, Cysteine only and CV&[Au$_{25}$(Cys)$_{18}$] of *E. coli* in white light (n=6 independent samples). Bacteria inoculated materials were exposed to white light with an intensity from 200 to 429 lux. Data presented as mean ± SD. ᵃCFU mL$^{-1}$ indicates Colony forming unit per mL. Blue star represents below detection limit: <10$^2$ CFU mL$^{-1}$.

reduction in the number of viable bacteria was not statistically significant on the samples with only CV or [Au$_{25}$(Cys)$_{18}$] alone or CV&[Au$_{25}$(Cys)$_{18}$] (*T*-test: *P* > 0.1), while a significant reduction in the numbers of viable bacteria was observed on the sample containing cysteine only (0.7 log reduction, *T*-test: *P* < 0.1). After 24 h exposure to white light, the polymer samples containing either CV or [Au$_{25}$(Cys)$_{18}$] alone did not show any bactericidal activity compared to the control, and the sample with cysteine only did not show any increase in bactericidal activity compared to that in the dark (*T*-test: *P* > 0.1). However, bactericidal activity of the polymer with CV&[Au$_{25}$(Cys)$_{18}$] was significantly enhanced after 24 h exposure of white light compared to the control, showing a 2.8-log reduction in the number of viable bacteria (*T*-test: *P* < 0.01).

*E. coli* was more resistant than *S. aureus* requiring a longer exposure time of white light to achieve a significant reduction in the number of viable bacteria. We attribute this to differences in

the cell wall, where *E. coli*—a Gram-negative bacterium—contains a double membrane structure compared to *S. aureus*—a Gram-positive bacterium—which contains only a single membrane barrier[41]. The outer membrane of Gram-negative bacterium decreases molecular penetration, and is often responsible for increased resistance to antibacterial agents[41,42].

It is known that the photobactericidal mechanism of CV starts with the absorption of light, resulting in an excited singlet state which has paired electron spin[22,28]. This either returns to the ground state or transforms to a triplet state, which has unpaired electron spin[22,28]. Molecules in the triplet state undergo two photochemical reaction processes known as Type-I and -II. For Type-I, redox reactions occur with their environment resulting in ROS generation[22,28]. For Type-II, the energy is transferred to ground state molecular oxygen, resulting in the transformation from triplet oxygen ($^3O_2$) to singlet oxygen ($^1O_2$)[22,28]. These ROS and $^1O_2$ result in a multi-site attack on bacteria resulting in cell

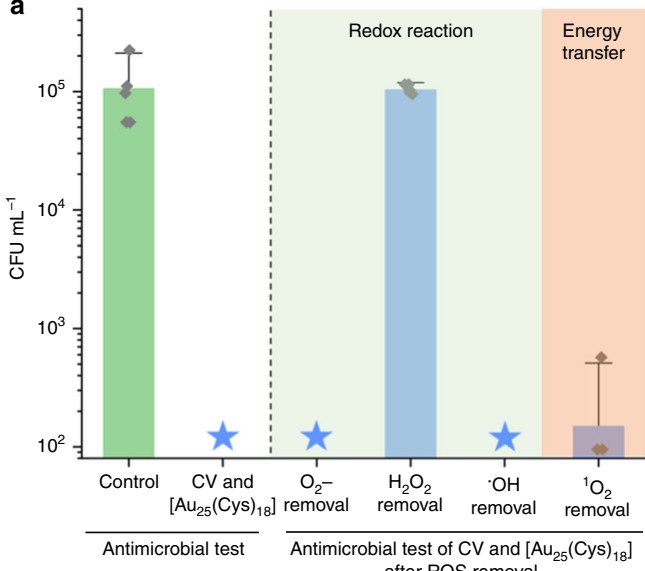

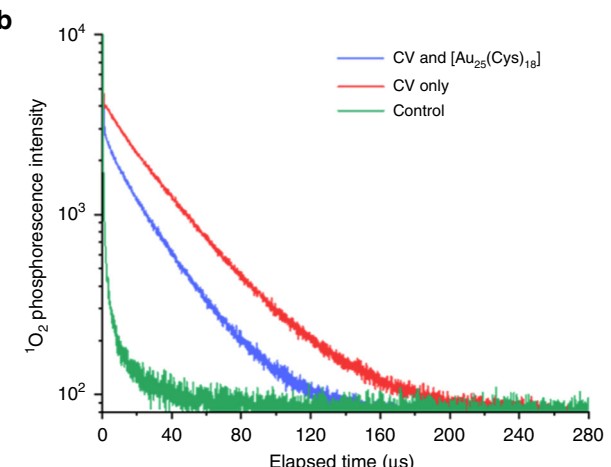

**Fig. 4 Characterisation of reactive oxygen species (ROS) induced by CV only and CV&[Au₂₅(Cys)₁₈].** a Bactericidal activity of CV&[Au$_{25}$(Cys)$_{18}$] after removal of reactive oxygen species (ROS) containing superoxide (O$_2^-$), hydrogen peroxide (H$_2$O$_2$), and singlet oxygen ($^1$O$_2$). CV&[Au$_{25}$(Cys)$_{18}$] samples were exposed to white light source with an intensity of 312 lux on average for 6 h (n=6 independent samples). CV indicates crystal violet. b Time-resolved $^1$O$_2$ phosphorescence decay on control, CV only, and [CV&Au$_{25}$(Cys)$_{18}$] ($\lambda_{Ex}$ = 532 nm, $\lambda_{Em}$ = 1270 nm). The phosphorescence decay was analysed using PicoQuant Fluofit software. Data in (a) presented as mean ± SD. Blue star represents below detection limit: <10$^2$ CFU mL$^{-1}$.

death[26]. The polymer containing CV only did not show any photobactericidal activity herein, upon exposure to low intensity white light. However, the addition of [Au$_{25}$(Cys)$_{18}$] to CV resulted in potent photobactericidal activity under identical conditions.

To determine the species responsible for the bactericidal effect observed in our CV&[Au$_{25}$(Cys)$_{18}$] system, ROS scavenger/ quencher assays and $^1$O$_2$ phosphorescence measurements were carried out. In the ROS scavenger/quencher assay, photo-bactericidal activity was measured in the presence of scavengers; superoxide dismutase (SOD) for superoxide radicals (O$_2^-$), cata-lase for hydrogen peroxide (H$_2$O$_2$), mannitol for hydroxyl radicals (•OH) and L-histidine for singlet oxygen species ($^1$O$_2$)[29,43–45]. As shown Fig. 4a, the polymer with CV&[Au$_{25}$(Cys)$_{18}$] exhibited

potent photobactericidal activity against *S. aureus* with >3.3-log reduction compared to the control after 6 h exposure of 312 lux white light without addition of ROS scavenger/quencher. When L-histidine ($^1$O$_2$ quencher) was added, the photobactericidal activity of the polymer was slightly reduced, but retained strong bacter-icidal activity with a 2.9 log reduction in the number of viable bacteria. O$_2^-$ and •OH scavenging by SOD and mannitol respectively did not cause any significant change in photo-bactericidal activity. However, the photobactericidal activity decreased from >3.3 to 0.02-log reduction when catalase (H$_2$O$_2$ scavenger) was added, indicating that the potent photobactericidal activity of CV&[Au$_{25}$(Cys)$_{18}$] is mainly due to H$_2$O$_2$ formation. Although the $^1$O$_2$ quencher assay showed that $^1$O$_2$ was produced by CV&[Au$_{25}$(Cys)$_{18}$] and gave some bacterial kill, it was not clear if the addition of [Au$_{25}$(Cys)$_{18}$] reinforced the Type-II pathway. Thus, to determine if Type-II was important, time-resolved near-infrared (TRNIR) spectroscopy was employed. A laser source with a wavelength of 532 nm was used for light illumination on the CV&[Au$_{25}$(Cys)$_{18}$] treated polymers and $^1$O$_2$ phosphorescence was measured at a wavelength of ~1270 nm[46,47]. Figure 4b shows the $^1$O$_2$ phosphorescence decay of the control, silicone with CV only and silicone with CV&[Au$_{25}$(Cys)$_{18}$] at a wavelength of 1270 nm. Upon laser illumination, the $^1$O$_2$ phosphorescence signal was observed on both the polymer with CV only and the polymer with CV&[Au$_{25}$(Cys)$_{18}$]. Interestingly, the lifetime of the $^1$O$_2$ phos-phorescence signal produced from the polymer with CV only (~33.8 µs) was longer than the polymer with CV&[Au$_{25}$(Cys)$_{18}$] (~25.5 µs). This indicates that the addition of [Au$_{25}$(Cys)$_{18}$] to CV appears to attenuate the Type-II pathway.

To understand the photoreaction mechanism of the polymer sample containing CV&[Au$_{25}$(Cys)$_{18}$], photocurrent measure-ments, steady state and time-resolved photoluminescence (PL) spectroscopies were employed. Transient photocurrent responses of CV only and CV&[Au$_{25}$(Cys)$_{18}$] treated polymers were mea-sured under several on-off cycles of white light irradiation. As shown in Fig. 5a, a greater rise in photocurrent of the polymers containing CV&[Au$_{25}$(Cys)$_{18}$] compared to CV alone was observed, indicating a higher CV concentration and/or a higher separation efficiency of electron–hole pairs. Figure 5b shows PL spectra of the samples with CV only and CV&[Au$_{25}$(Cys)$_{18}$] from 600 to 800 nm. The intensity of the PL spectrum is a direct measurement on recombination rate of electron–hole pair. The higher the peak intensity of the spectrum, the easier the recom-bination of the electron–hole pair. A PL peak of the polymer sample with CV only was observed at ~675 nm, and the PL peak of the sample with CV&[Au$_{25}$(Cys)$_{18}$] was lower than that of CV only indicating that the recombination rate of the photogenerated electron–hole pairs on CV&[Au$_{25}$(Cys)$_{18}$] is slower than that of CV only. This indicates that additional encapsulation of [Au$_{25}$(Cys)$_{18}$] into CV-treated polymer improves the photoelec-trochemical properties of the sample. Figure 5c shows time-resolved PL decay of samples containing CV only or CV& [Au$_{25}$(Cys)$_{18}$]. The PL decay was measured at a wavelength of 650 nm upon excitation by a laser source with a wavelength of 574 nm. Compared to the CV only polymer, the PL life time of the polymer sample with CV&[Au$_{25}$(Cys)$_{18}$] was shorter. This was attributed to electron transfer from CV to [Au$_{25}$(Cys)$_{18}$][48,49]. It is suggested that [Au$_{25}$(Cys)$_{18}$] on the CV-treated surface acts as an electron acceptor, resulting in better electron–hole pair separation, a reduction in recombination, and enhanced photo-bactericidal properties[48,49].

In order to determine the band offset of the CV and [Au$_{25}$(Cys)$_{18}$], XPS and UV–Vis spectroscopy were used, with the results shown in Fig. 6. For XPS analysis, the adventitious C 1s peak at 284.8 eV was used as a reference for charge correction. Band gap energies ($E_G$) of CV and [Au$_{25}$(Cys)$_{18}$] were 1.95 and 1.25 eV,

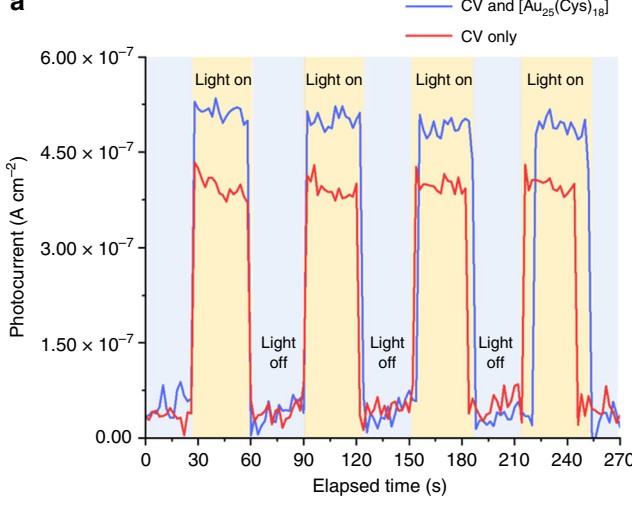

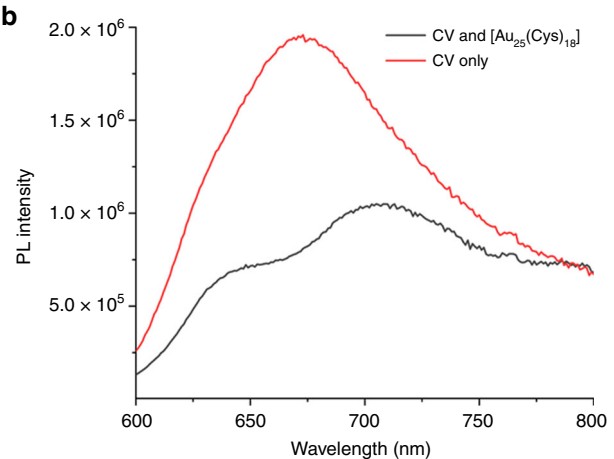

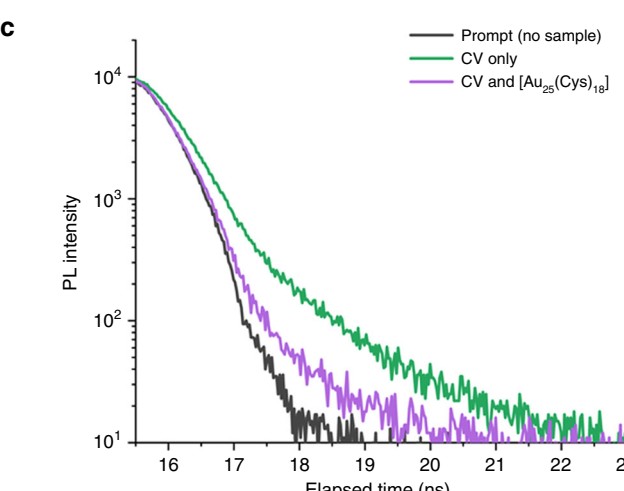

**Fig. 5 Photoreaction characterisation of CV only and CV&[Au$_{25}$(Cys)$_{18}$].
a** Transient photocurrent responses for CV only and CV&[Au$_{25}$(Cys)$_{18}$] under white light. CV indicates crystal violet. **b** Photoluminescence (PL) spectra of CV only and CV&[Au$_{25}$(Cys)$_{18}$] in wavelength of 600 to 800 nm ($\lambda_{Ex} = 574$ nm). **c** Time-resolved photoluminescence (PL) decay of CV only and CV&[Au$_{25}$(Cys)$_{18}$] ($\lambda_{Ex} = 574$ nm, $\lambda_{Em} = 650$ nm).

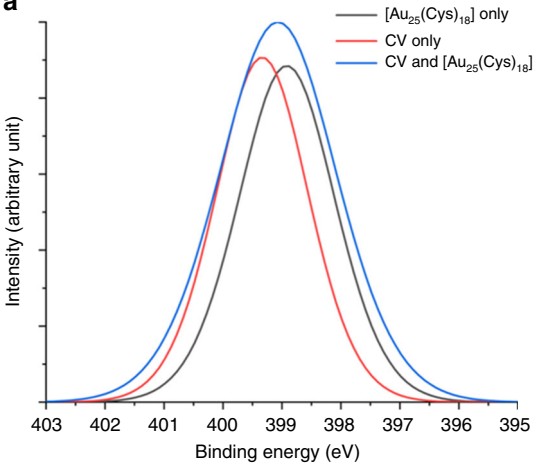

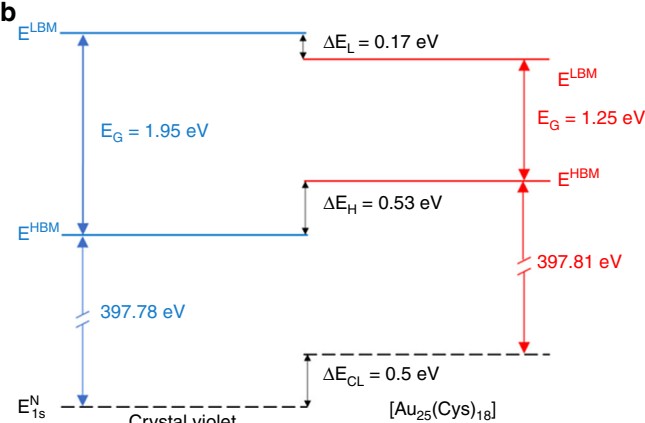

**Fig. 6 Band alignment of crystal violet and [Au$_{25}$(Cys)$_{18}$] from XPS. a** N 1s spectra taken from CV only, [Au$_{25}$(Cys$_{18}$)] only and CV&[Au$_{25}$(Cys)$_{18}$)]. Experimental data shown as blue line are fitted with the peak shapes derived from phase-pure crystal violet (red line) and [Au$_{25}$(Cys)$_{18}$] (black line). **b** XPS band alignment between crystal violet and [Au$_{25}$(Cys)$_{18}$]. CV indicates crystal violet. $E_{HBM}$ is homo band maximum energy. $\Delta E_{H}$ is the energy offset between CV $E_{HBM}$ and [Au$_{25}$(Cys)$_{18}$] $E_{HBM}$. $E_{LBM}$ is lumo band maximum energy. $\Delta E_{L}$ is the energy offset between CV $E_{LBM}$ and [Au$_{25}$(Cys)$_{18}$] $E_{LBM}$. $\Delta E_{CL}$ is the core level offset between the N 1s core levels.

respectively (Supplementary Fig. 6a and d) and homo band maximum energies ($E^{HBM}$) of CV and [Au$_{25}$(Cys)$_{18}$] were 1.6 and 1.07 eV, respectively (Supplementary Fig. 6b, e). A difference of binding energy between N 1s core level (CL) and homo band was investigated in CV only and [Au$_{25}$(Cys)$_{18}$] only samples (Supplementary Fig. 6c, f). The same functions were used to peak fit N 1s peaks in CV&[Au$_{25}$(Cys)$_{18}$] to determine the binding energy (Fig. 6a). The N 1s CL environments can be considered equivalent in these materials, and can therefore be used as a reference point to determine the band alignment. As shown in Fig. 6b, the difference of binding energy between N 1s CL and $E^{HBM}$ was 397.78 eV for CV and 397.81 eV for [Au$_{25}$(Cys)$_{18}$]. The $E^{HBM}$ of [Au$_{25}$(Cys)$_{18}$] was 0.53 eV higher than that of CV while the lumo band maximum energy ($E^{LBM}$) of CV was 0.17 eV higher than that of [Au$_{25}$(Cys)$_{18}$], indicating the formation of a straddling (type I) band alignment at the interface between CV and [Au$_{25}$(Cys)$_{18}$][50,51]. This alignment suggests, that upon white light illumination, photogenerated electrons can flow from CV to [Au$_{25}$(Cys)$_{18}$], resulting in electron accumulation in [Au$_{25}$(Cys)$_{18}$]; in agreement with time-resolved PL measurements (Fig. 5c).

For the widespread use of photobactericidal coatings in hospitals and other indoor facilities, the light source should be carefully considered. Firstly, the light source should not produce

**Table 1 Recommended guideline for lighting in UK healthcare buildings.**

| Location | Illumination (lux) |
|---|---|
| Reception and enquiry desk | 300–500 |
| Desk in nurse station | 300 |
| Bedded (ward) area | |
| Patient reading | 300 |
| General nursing care | 300 |
| Examination and treatment | 1000 |
| Operating theatre | |
| Working plane | 500–1000 |
| Operating table | 15,000–30,000 |
| Accident and Emergency (A&E) unit | |
| Working plane | 500–1000 |
| Couch | 15,000–90,000 |
| Dentistry unit | 500–1000 |
| Maternity unit | 300–1000 |
| Sterilising and disinfection unit | 300 |
| Diagnostic equipment room | 500 |

The guideline was sourced from lighting guide 2 - hospitals and health care buildings[61].

an adverse effect on hospital staff and patients. Secondly, because light sources typically range from 300 (corridors, rooms) to 90,000 lux (operating theatre) in hospitals and healthcare facilities (Table 1), potent photobactericidal activity should be present at various lighting conditions. Titanium dioxide ($TiO_2$) and zinc oxide (ZnO) nanoparticles are the most extensively studied photocatalysts, and they exhibit a broad photobactericidal spectrum[52–54]. However, their bactericidal activity is negligible under white light sources, which are commonly used in healthcare facilities, because $TiO_2$ and ZnO nanoparticles are UV-active photocatalysts, and the portion of UV light present in indoor lighting is typically extremely low[52]. To produce enhanced indoor photocatalysts, $TiO_2$ based nanocomposites using Ag NPs and Au NPs, carbon nanotube (CNT), and graphene oxide (GO) etc have been studied[53–55]. Although some of the composites exhibited bactericidal activity under a white light source, such studies typically employed an intense light source of >1300 lux (0.2 mW $cm^{-2}$) indicating lack of feasibility for use in healthcare facilities (Table 1)[2,53–58]. In recent years, it was reported that TBO, CV, and MB treated polymer surfaces exhibit photobactericidal activity and the addition of 2 nm Au NPs or 3 nm ZnO NPs within the treated surface results in enhanced photobactericidal activity[24,26,29,59]. However, intense white light of >1000 lux (>0.15 mW $cm^{-2}$) or a ~600 nm laser source had to be employed to achieve potent bacterial kill, indicating that such polymers can only be used in healthcare environments with extremely bright lighting, such as operating theatres and A&E examination units (Table 1)[24,26,29,59,60]. Our study showed that the addition of $[Au_{25}(Cys)_{18}]$ into a CV impregnated polymer significantly enhances its photobactericidal activity, showing potency under low intensity white light ranging from 200 to 429 lux (0.03 to 0.06 mW $cm^{-2}$), which is >3 times lower light levels than previous studies[24,26,29,53–55,59,60]. Moreover, this dye-cluster combination can be applied to a wide range of devices based on polymer such as endotracheal tubing, keyboard cover, catheter, screen cover, tablet and phone covers.

The polymer encapsulated with CV&$[Au_{25}(Cys)_{18}]$ displayed potent bactericidal activity against *S. aureus* and *E. coli*, which are both associated with hospital acquired infections (HAIs). An illumination with a white light source (312 lux) of similar intensity to that commonly found in healthcare facilities resulted in a significant enhancement of the bactericidal activity of the polymer with >3.3 and 2.8 log reductions for *S. aureus* and *E. coli*

respectively after 6 and 24 h of exposure to white light. Our studies of the photobactericidal mechanism showed that upon white light illumination, photogenerated electrons in CV likely flow into $[Au_{25}(Cys)_{18}]$ (Fig. 6). This indicated that redox reaction was induced from $[Au_{25}(Cys)_{18}]$. Chemical scavenger and TRNIR spectroscopy studies showed that the Type-II pathway for forming $^1O_2$ was suppressed in the CV&$[Au_{25}(Cys)_{18}]$ system, and that the redox reaction to form $H_2O_2$ was promoted. The photobactericidal polymer containing CV&$[Au_{25}(Cys)_{18}]$, which we report herein is a promising candidate for use in healthcare environments to prevent the spread of HAIs because its potent photobactericidal activity at ambient light levels (Table 1).

## Methods

**Synthesis of gold nanocluster ($[Au_{25}(Cys)_{18}]$).** Supplementary Fig. 1 shows a set-up of microfluidic segmented flow system to synthesize $[Au_{25}(Cys)_{18}]$. It consists of a tube-in-tube membrane contactor, and a coil flow inverter reactor. The tube-in-tube contactor consists of polytetrafluoroethylene (PTFE) tubing (inner diameter: 3.2 mm, VICI Jour), and Teflon AF-2400 tubing (inner diameter: 0.8 mm, Cambridge Reactor Design Ltd) located in PTFE tubing. CO (BOC) was dissolved into heptane in the tube-in-tube contactor and then the mixture was saturated under 500 kPa. 21.8 mg of cysteine and 3 mL of tetrachloauric acid stock solution (59.5 mM, Sigma-Aldrich) were mixed in de-ionized (DI) water to form 40 mL of the thiolate-Au (I) complex solution (3 mM), the pH of the solution was maintained at 11.6 using sodium hydroxide (NaOH, 2 N, Sigma-Aldrich). The complex was mixed with CO-saturated heptane and then it passed through the coiled flow inverter at 80 °C. The coiled flow inverter reactor consists of 100 coils of fluorinated ethylene propylene (FEP) tubing (inner diameter: 1 mm, VICI Jour). The diameter of the coil was about 1 cm and 90° bend was applied at every five coils. After the reactor, the solution was mixed with nitrogen gas in the collection flask. Through the microfluidic segmented flow system, $[Au_{25}(Cys)_{18}]$ clusters were synthesized within about 3 minutes of residence time. $[Au_{25}(Cys)_{18}]$ solution was placed at room temperature overnight. After that, an organic layer which is formed by heptane on the top of the solution was removed.

**TEM and UV–Vis absorbance spectroscopy.** In order to determine the size and morphology of the cluster, transmission electron microscopy (TEM, JEOL 2100, JEOL Ltd.) was used. A droplet of the cluster solution was inoculated on the TEM grid and dried in air. The TEM image of $[Au_{25}(Cys)_{18}]$ was taken at an accelerating voltage of 120 kV. UV–Vis absorption spectrum of $[Au_{25}(Cys)_{18}]$ solution was measured using a UV–Vis spectrometer with mini halogen light source (USB 2000 +UV-Vis-ES, Ocean Optic Inc.).

**Electrospray ionization-mass spectrometry.** To characterise $[Au_{25}(Cys)_{18}]$, the dried nanoclusters were mixed with DI water containing 0.01 μM cesium acetate, and then the solution was infused into an electrospray ionization-mass spectrometer (ESI-MS, Q-TOF 6510, Agilent Technology) at a flow rate of 20 μL min$^{-1}$. The electrospray ionization (ESI) was operated in negative mode, and heated nitrogen (the drying gas) was supplied into the spectrometer at a flow rate of 5 L min$^{-1}$. The negatively charged ions were analysed by mass spectrometer (MS).

**Concentration of gold nanocluster ($[Au_{25}(Cys)_{18}]$).** The concentration of synthesized $[Au_{25}(Cys)_{18}]$ was calculated by Eq. (1):

$$C = \frac{A}{\varepsilon \ell} \tag{1}$$

where C and A indicate the concentration (mM) of $[Au_{25}(Cys)_{18}]$ clusters and the absorbance of the cluster solution at 670 nm, respectively, $\varepsilon$ and $\ell$ represent molar absorption coefficient at 670 nm and path length through the sample, respectively.

**Preparation of photobactericidal surface.** In a mixture of acetone (140 mL), $[Au_{25}(Cys)_{18}]$ clusters (28 mL), distilled (DI) water (112 mL) or acetone (140 mL) and DI water (140 mL), 224 mg of crystal violet (CV, Sigma-Aldrich) was dissolved at a concentration of 800 ppm. Silicone sheet (4.5 cm × 7.0 cm) was immersed in 280 mL of was dissolved for 24 h. The sheet was collected from the solution, washed by DI water two times to remove unbound materials, and then air dried for 24 h in a dark room. After that, the sheet was cut into small pieces (1.5 cm × 1.5 cm).

**Characterisation of photobactericidal surface.** UV–Vis absorbance spectra of the control, and treated silicone samples were measured by UV–Vis spectrometer (Lambda 25, PerkinElmer Inc.), and the absorbance was measured in a range of 400 −1000 nm.

To determine the CV diffusion in silicone, the polymer was immersed in CV solution for 0.5, 1, 5 and 24 h, and it was sliced into 30 μm thickness using a microtome. The sliced sample was mounted onto a glass slide and then its side section

was imaged using fluorescence microscopy (Olympus BX63, Tokyo, Japan) with TRITC filter (excitation/emission wavelengths 545 nm/620 nm). The samples were also photographed by optical microscopy, and then the image was analysed using ImageJ (http://imagej.nih.gov/ij/) to obtain the cross-sectional intensity profiles.

**X-ray photoelectron spectroscopy (XPS)**. XPS depth profile was employed to determine the existence of $[Au_{25}(Cys)_{18}]$ within the treated polymer matrix and at the polymer surface. The binding energies of Au atom were determined after 0, 200 and 400 s of sputtering.

**X-ray fluorescence (XRF) analysis**. XRF analysis was employed to determine a change of CV concentration after additional encapsulation of $[Au_{25}(Cys)_{18}]$ into CV-treated polymer. The treated polymer (1.5 cm × 1.5 cm) was placed into pot with colourless substrate and then it was located into XRF spectrometer (15watt Epsilon 4, Malvern Panalytical Ltd, UK).

**Bactericidal test**. The bactericidal activity of control and the treated polymers were tested against *Staphylococcus aureus* 8325-4 and *Escherichia coli* ATCC 25922 under dark and white light conditions. *S. aureus* and *E. coli* were stored in brain heart infusion broth (BHI, Oxoid Ltd., Hampshire, England, UK) with 20% gly-cerol at −70 °C, respectively and then they were propagated on either mannitol salt agar (Oxoid Ltd.) or MacConkey agar (Oxoid Ltd.) at 37 °C, respectively. One of the bacterial colonies grown on the agar was inoculated into BHI broth and cul-tured at 37 °C with 200 rpm shaking. After 18 h of culture, the bacteria were harvested through centrifugation (20 °C, 2795 × g for 5 min), washed using 10 mL of PBS to remove BHI broth and then centrifuged to get the bacteria resuspended into 10 mL of PBS. The washed bacterial suspension was 10000-fold diluted to obtain ~$10^5$ CFU mL$^{-1}$. As shown in Supplementary Fig. 7, 25 μL of the bacterial suspension was inoculated onto the control and the treated sample surface and then the samples were located in a colourless petri dish containing wet filter paper to obtain constant humidity. After that, the samples were exposed to white light (Osram L58W/865Lumilux, Munich, German) while a same set of samples was kept in a dark room, and then the samples were placed in 450 μL of PBS and vortexed for 1 min to wash out bacteria from the sample into the PBS. After a serial dilution, *S. aureus* and *E. coli* suspensions (each 100 μL) were plated onto mannitol salt agar and MaConkey agar, respectively and cultured at 37 °C for 24 h. The bacterial colonies which were grown in the agar were counted. Each experiment contained two technical replicates and the experiment was reproduced three times.

**Detection of superoxide radical, hydrogen peroxide, hydroxyl radical and singlet oxygen species**. *S. aureus* 8425-4 and catalase (Sigma-Aldrich), L-histidine (Sigma-Aldrich), mannitol (Sigma-Aldrich) and superoxide dismutase (SOD, Sigma-Aldrich) which are scavengers and quenchers of ROS were used. Catalase was used at a concentration of 6–14 unit mL$^{-1}$ in bacterial solution to remove hydrogen peroxide ($H_2O_2$). SOD was used at a concentration of ~20 unit mL$^{-1}$ in bacterial suspension to remove superoxide ($O_2^-$) radicals. Mannitol was used at a concentration of 82 mM in bacterial solution to eliminate hydroxyl radicals (•OH). L-histidine was used at a concentration of 2 mM in bacterial solution as singlet oxygen ($^1O_2$) quencher. The catalase and SOD scavengers which are enzyme act extracellularly since they are large molecules. Thus, the catalase intercepts the hydrogen peroxide coming from the surface. Twenty-five microlitres of the bac-terial suspension containing ROS scavenger or quencher was inoculated onto antimicrobial sample, the samples were located on to a colourless petri dish con-taining wet filter paper, and then they were was exposed to white light for 6 h. After that, samples were located in 450 μL of PBS and vortexed for 1 min to wash out bacteria from the sample to PBS. After a serial dilution, 100 μL of bacterial sus-pension was plated onto mannitol salt agar and cultured at 37 °C for 24 h. The bacterial colonies which were grown in the agar were counted. Each experiment contained two technical replicates and the experiment was reproduced three times.

**Detection of time-resolved singlet oxygen phosphorescence**. In order to detect singlet oxygen ($^1O_2$) phosphorescence, a near-infrared sensitive thermoelec-trically cooled photomultiplier was employed. The silicone sample was placed onto a slide glass and irradiated by Nd:YAG laser operating at 532 nm. A PC-mounted multiscaler board with pre-amplifier (MSA-300, Becker-Hickl) was used as the photon counting system to measure singlet oxygen phosphorescence at a wavelength of ~1270 nm. After the measurement, the data was analysed using FluoFit software (PicoQuant GmbH).

**Photoelectrochemical measurement and PL spectroscopy**. Photocurrent mea-surements were performed using Metrohm Autolab (PGSTAT302N, Utrecht, Netherlands) with a three-electrode system with an external source of white light. Ag/AgCl reference electrode, and platinum sheet as a counter electrode. The photobactericidal polymer (CV only and CV&$[Au_{25}(Cys_{18})]$) served as the working electrode. The electrolyte used in the system was 0.1 M $Na_2SO_4$.

PL spectra of samples were measured in a wavelength of 600 to 800 nm using steady state PL spectrometer (FluoroMax, Horiba Scientific, Kyoto, Japan). The excitation wavelength was ~574 nm.

**Time-resolved PL spectroscopy**. Time-resolved PL spectroscopy, for timescales up to 100 ns (~24.4 ps resolution), was performed using a time-correlated single photo counting (TCSPC) apparatus (DeltaFlex, Horiba Scientific). Pulsed 574 nm excitation (1 Mz repetition rate, <1.6 ns pulse width) was generated by a laser diode (NanoLED-570), and the fluorescence was detected at wavelengths at 650 nm (Picosecond Photon Detection Module, PPD-900, Horiba Scientific, Kyoto, Japan).

**Zeta potential measurement**. Zeta potential of the $[Au_{25}(Cys)_{18}]$ in the mixture of water and acetone were measured by Delsa zeta potential anlyzer (DelsaMax-Pro, Beckman Coulter) with the available flow cell system in batch mode at 22 °C.

**Measurement of band offset**. XPS was employed to determine the band offset of CV and $[Au_{25}(Cys)_{18}]$. The energy difference between a CL and homo band maximum (HBM) for individual materials was measured and then the difference between CLs of CV and $[Au_{25}(Cys)_{18}]$ was measured. Subsequently, the band alignment and energy offset were determined by Eq. (2):

$$\Delta E_{HBM} = \left(E_{CL}^{Au\,cluster} - E_{HBM}^{Au\,cluster}\right) - \left(E_{CL}^{CV} - E_{HBM}^{CV}\right) - \Delta E_{CL} \qquad (2)$$

where $E_{CL}^{Au\,cluster}$ and $E_{CL}^{CV}$ are the energy of $[Au_{25}(Cys)_{18}]$ and CV core levels, respectively and $E_{HBM}^{Au\,cluster}$ and $E_{HBM}^{CV}$ are the energy of $[Au_{25}(Cys)_{18}]$ and CV HBMs, respectively. $\Delta E_{CL}$ is $E_{CL}^{Au\,cluster} - E_{CL}^{CV}$ and $\Delta E_{HBM}$ is $E_{HBM}^{Au\,cluster} - E_{HBM}^{CV}$. Additionally, to determine the peaks' position of XPS spectra precisely, Shirley background and Gaussian-Lorentzian profiles were used.

**Statistical analysis**. Statistical *T*-test of results was calculated by Excel software (Microsoft corporation, NM, USA).

**Reporting summary**. Further information on research design is available in the Nature Research Reporting Summary linked to this article.

## Data availability
The data presented in this manuscript are available from the corresponding author upon reasonable request. The source data underlying Figs. 1a, c, d, 2b, 4a, b, 5a–c, 6a, and Supplementary Figs. 2, 3b, c, 6a–f are provided as a Source Data file.

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

## Acknowledgements

The authors thank the EPSRC for financial support (EP/M015157/1) through the Manufacturing Advanced Functional Materials (MAFuMa) scheme. H.H. acknowledges the financial support from a UCL-CSC scholarship.

## Author contributions

G.B.H. designed and performed the experiments and wrote and revised the manuscript. E.A., A.G. and I.P.P. designed and supervised experiments and revised the manuscript. H.H. and G.W. synthesised and characterised [Au$_{25}$(Cys)$_{18}$]. K.K. and H.D.T. advised on the synthesis and characterisation of [Au$_{25}$(Cys)$_{18}$]. A.M.A. carried out XPS analysis. A.J.M. advised and carried out mechanistic studies. A.K. carried out time-resolved PL measurements. L.M.H. carried out TRNIR microscopy. J.S. carried out XPS, photocurrent measurements and mechanistic studies.

## Competing interests

The authors declare no competing interests.
