## [Peer Review File · Nature Communications]

Reviewers' Comments:

Reviewer #1:

Remarks to the Author:

Manuscript No: NCOMMS-19-08856-T

Manuscript Title: Photobactericidal activity of crystal violet encapsulated silicone activated by thiolated gold nanoclusters at low flux levels of white light.

In this study, the authors evaluate the photobactericidal effect of a polymer containing crystal violet (CV) and gold nanocluster Au₂₅(Cys) activated at a low flux levels of white light.

Comments:

The manuscript is on an interesting topic with fundamental and applied perspective: how to prevent and/or to stop the development of hospital acquired infections. However, there are some aspects that need to be improved/clarified.

As the authors stated, it is the first time that a gold cluster enhance the antimicrobial activity of a light activated dye is reported, however, no mechanism explaining this synergistic effect is proposed. The authors only tested the effect of different scavengers on the photoinactivation efficiency of the bacterial strain, concluding that H₂O₂ is the main intervenient in the process. According to the literature, CV act mainly through reaction Type II. In this study using the Au/CV polymer seems that other alternative pathway occurs, but it is difficult to understand which mechanism is behind the process of bacterial inactivation. An explanation about the alternative photoinactivation pathway should be added to the manuscript. Moreover, no indication about the CV concentration in the polymer with CV and in the polymer with CV/Au is presented. However, the Figure 2c of the supplementary information shows that the concentration of CV in the polymer with only CV is much lower than that in the polymer with CV/Au, at least 2 times lower. This can explain, at least, in part, the difference in bacterial inactivation by the two polymers. The information about the CV concentration in both polymers is essential to interpret the results and should be added to the manuscript and discussed.

The experiments were done with a Gram positive bacterium, but it is well known that Gram negative bacteria are more difficult to inactivate by photodynamic therapy than Gram positive ones. No explanation is presented about why the authors choose a Gram positive bacterium to test the new material.

There is no indication about the dark period of incubation before the phototreatment. The treatment was done on a solid surface, but the results are expressed by mL. Which was the bacterial decrease by cm² of the polymer?

The light intensity used in the study is expressed in lux, but to compare with other studies already described in the literature the intensity should be also expressed in mw/cm².

How was the bacterial inoculum spread at the surface of the materials? There is no indication in the supplementary information.

No indication about the number of independent assays done is indicated. The number of replicates per dilution is also not presented.

The detection limit of the method used to determine the bacterial concentration is too high < 10² CFU/mL. Is it correct?

The authors stated that "The ROS and 1O₂, produce", but the 1O₂ is a ROS.

Throughout the manuscript in many cases a "," is used before the "and", in these cases the "," should be removed.

I can not understand the Figure 3a of the supplementary information.

According to Figure 3b of the supplementary information, around 10% of the used light is UV, but in the manuscript it is indicated that the experiments were done with white light.

Adelaide Almeida

Reviewer #2:

Remarks to the Author:

Photobactericidal activity of crystal violet encapsulated silicone activated by thiolated gold nanoclusters at low flux levels of white light

The above title manuscript is very preliminary work not worth publishing Nature Communication.

General comments:

What is the novelty of the work as this kind of work has already been published.

1. Naik AJ, Ismail S, Kay C, Wilson M, Parkin IP. Antimicrobial activity of polyurethane embedded with methylene blue, toluidene blue and gold nanoparticles against Staphylococcus aureus; illuminated with white light. Materials Chemistry and Physics. 2011 Sep 15;129(1-2):446-50.
2. Perni S, Piccirillo C, Pratten J, Prokopovich P, Chrzanowski W, Parkin IP, Wilson M. The antimicrobial properties of light-activated polymers containing methylene blue and gold nanoparticles. Biomaterials. 2009 Jan 1;30(1):89-93.

What is the rationale behind this work?, Not clear.

What is the mechanism of killing? The author stated on line number 67 that "gold cluster could promote a greater synergistic effect " if author was checking synergistic effect then they should go for checker board assay.

Photobactericidal mechanism is due to which compound? There is no killing after light activation in Au₂₅(Cys)₁₀, CV and Cysteine then what happen to CV&Au₂₅(Cys)₁₀? What is the mechanism behind the Photobactericidal activity of combined CV&Au₂₅(Cys)₁₀?

Overall this manuscript lack novelty as well as they fail to justify their own work.

Reviewers' comments:

Reviewer #1 (Remarks to the Author):

Manuscript No: NCOMMS-19-08856-T

Manuscript Title: Photobactericidal activity of crystal violet encapsulated silicone activated by thiolated gold nanoclusters at low flux levels of white light.

In this study, the authors evaluate the photobactericidal effect of a polymer containing crystal violet (CV) and gold nanocluster Au₂₅(Cys)₁₈ activated at a low flux levels of white light.

Comments:

The manuscript is on an interesting topic with fundamental and applied perspective: how to prevent and/or to stop the development of hospital acquired infections. However, there are some aspects that need to be improved/clarified.

Response

We appreciate the comments and suggestions made by reviewer 1. We have addressed them in our revised manuscript as specified below

1. As the authors stated, it is the first time that a gold cluster enhance the antimicrobial activity of a light activated dye is reported, however, no mechanism explaining this synergistic effect is proposed. The authors only tested the effect of different scavengers on the photoinactivation efficiency of the bacterial strain, concluding that H₂O₂ is the main intervenient in the process. According to the literature, CV act mainly through reaction Type II. In this study using the Au/CV polymer seems that other alternative pathway occurs, but it is difficult to understand which mechanism is behind the process of bacterial inactivation.

Response

So far, it was reported that CV induces superoxide radical and singlet oxygen through reaction Type-I and II. However, ROS identities induced from CV was not extensively determined. In our experiment, ROS scavenger/quencher assay and ¹O₂ phosphorescence measurement showed that the redox reaction was enhanced and the Type-II pathway was decreased, but it was not explained how the redox reaction was enhanced. For better understanding, additional experiments were performed, and the results showed that upon white light illumination, photogenerated electrons in crystal violet flows from crystal violet to the gold nanocluster. As a result, excessive electron accumulation in the cluster cause electron transfer to the environment and promotes redox reaction. The results and discussion of the photoreaction mechanism and Materials&Methods were added into manuscript (Page 9 to 11, Figure 5 and 6, and Supplementary Figure 5).

Information as below was added into the manuscript

“To understand the photoreaction mechanism of the polymer sample containing CV&[Au₂₅(Cys)₁₈], photocurrent measurements, steady state and time-resolved photoluminescence (PL) spectroscopies were employed. Transient photocurrent responses of CV only and CV&[Au₂₅(Cys)₁₈] treated polymers were measured under several on-off cycles of white light irradiation. As shown in Figure 5a, a greater rise in photocurrent of the polymers containing CV&[Au₂₅(Cys)₁₈] compared to CV alone was observed, indicating a higher separation efficiency of electron-hole pairs. Figure 5b shows PL spectra of the samples with CV only and CV&[Au₂₅(Cys)₁₈] from 600 to 800 nm. The intensity of the PL spectrum is a direct measurement on recombination rate of electron-hole pair. The higher the peak intensity of the spectrum, the easier the recombination of the electron-hole pair. A PL peak of the polymer sample with CV only was observed at ~675 nm, and the PL peak of the sample with CV&[Au₂₅(Cys)₁₈] was lower than that of CV only indicating that the recombination rate of the photogenerated electron-hole pairs on CV&[Au₂₅(Cys)₁₈] is slower than that of CV only. This indicates that additional encapsulation of [Au₂₅(Cys)₁₈] into CV treated polymer improves the photoelectrochemical properties of the sample. Figure 5c shows time-resolved PL decay of samples containing CV only or CV&[Au₂₅(Cys)₁₈]. The PL decay was measured at a wavelength of 650 nm upon excitation by a laser source with a wavelength of 574 nm. Compared to the CV only polymer, the PL life time of the polymer sample with CV&[Au₂₅(Cys)₁₈] was shorter. This was attributed to electron transfer from CV to [Au₂₅(Cys)₁₈].^{47,48} It is suggested that [Au₂₅(Cys)₁₈] on the CV treated surface acts as electron acceptor, resulting in better electron-hole pair separation, a reduction in recombination, and enhanced photobactericidal properties.^{47,48}

In order to determine the band offset of the crystal violet and [Au₂₅(Cys)₁₈], XPS and UV-Vis spectroscopy were used, with the results shown in Figure 6. For XPS analysis, the adventitious C 1s peak at 284.8 eV was used as a reference for charge correction. Band gap energies (E_G) of crystal violet and [Au₂₅(Cys)₁₈] were 1.95 and 1.25 eV, respectively (Supplementary Figures 6a and d) and homo band maximum energies (E_{HBM}) of crystal violet and [Au₂₅(Cys)₁₈] were 1.6 and 1.07 eV, respectively (Supplementary Figures 6b and e). A difference of binding energy between N 1s core level (CL) and homo band was investigated in CV only and [Au₂₅(Cys)₁₈] only samples (Supplementary Figures 6c and f). The same functions were used to peak fit N 1s peaks in CV&[Au₂₅(Cys)₁₈] to determine the binding energy (Figure 6a). As shown in Figure 6b, the difference of binding energy between N 1s CL and E_{HBM} was 397.78 eV for CV and 397.81 eV for [Au₂₅(Cys)₁₈]. The E_{HBM} of [Au₂₅(Cys)₁₈] was 0.53 eV higher than that of CV while the lumo band maximum energy (E_{LBM}) of crystal violet was 0.17 eV higher than that of [Au₂₅(Cys)₁₈], indicating the formation of a straddling (type I) band alignment at interface between CV and [Au₂₅(Cys)₁₈].^{49,50} This alignment suggests, that upon white light illumination, photogenerated electrons can flow from CV to [Au₂₅(Cys)₁₈], resulting in electron accumulation in [Au₂₅(Cys)₁₈]; in agreement with time-resolved PL measurements (Figure 5c).”

Figure 5 Photoreaction characterisation of CV only and CV&[Au₂₅(Cys)₁₈]. **a**, Transient photocurrent responses for CV only and CV&[Au₂₅(Cys)₁₈] under white light. **b**, photoluminescence (PL) spectra of CV only and CV&[Au₂₅(Cys)₁₈] in wavelength of 600 to 850 nm (λ_{Ex} = 574 nm). **c**, Time-resolved photoluminescence (PL) decay of CV only and CV&[Au₂₅(Cys)₁₈] (λ_{Ex} = 574 nm, λ_{Em} = 650 nm).

Figure 6 band alignment of crystal violet and $[\text{Au}_{25}(\text{Cys})_{18}]$ from XPS. **a**, N 1s spectra taken from CV only, $[\text{Au}_{25}(\text{Cys})_{18}]$ only and CV& $[\text{Au}_{25}(\text{Cys})_{18}]$. Experiment data shown as blue line are fitted with the peak shapes derived from phase-pure crystal violet (red line) and $[\text{Au}_{25}(\text{Cys})_{18}]$ (black line). **b**, XPS band alignment between crystal violet and $[\text{Au}_{25}(\text{Cys})_{18}]$. ΔE_{CL} is the core level offset between the N 1s core levels.

Supplementary Figure 6 | band gap, homo band and N 1s spectra from phase-pure crystal violet and $[\text{Au}_{25}(\text{Cys})_{18}]$. **a**, band gap, **b**, homo band and **c**, N 1s spectra for phase-pure crystal violet. **d**, band gap, **e**, homo band and **c**, N 1s spectra for phase-pure $[\text{Au}_{25}(\text{Cys})_{18}]$.

Materials and method

“Photoelectrochemical measurement and photoluminescence spectroscopy. Photocurrent measurements were performed using Metrohm Autolab (PGSTAT302N, Utrecht, Netherlands) with a three-electrode system with an external source of white light. Ag/AgCl reference electrode, and platinum sheet as a counter electrode. The photobactericidal polymer (CV only and CV& $[\text{Au}_{25}(\text{Cys})_{18}]$) served as the working electrode. The electrolyte used in the system was 0.1 M Na_2SO_4 .

Photoluminescence (PL) spectra of samples were measured in a wavelength of 600 – 800 nm using steady state PL spectrometer (FluoroMax, Horiba Scientific, Kyoto, Japan). The excitation wavelength was ~574 nm.

Time-resolved photoluminescence spectroscopy. Time-resolved photoluminescence spectroscopy, for timescales up to 100 ns (~24.4 ps resolution), was measured using a time-correlated single photo counting (TCSPC) apparatus (DeltaFlex, Horiba Scientific). Pulsed 574 nm excitation (1 Mz repetition rate, <1.6 ns pulse width) was generated by a laser diode (NanoLED-570), and the fluorescence was detected at wavelengths at 650 nm (Picosecond Photon Detection Module, PPD-900, Horiba Scientific, Kyoto, Japan).

Zeta potential measurement

Zeta potential of the $[\text{Au}_{25}(\text{Cys})_{18}]$ in the mixture of water and acetone were measured by Delsa zeta potential analyzer (DelsaMax-Pro, Beckman Coulter) with the available flow cell system in batch mode at 22 °C.

Measurement of band offset

XPS was employed to determine the band offset of crystal violet and $[\text{Au}_{25}(\text{Cys})_{18}]$ using the method proposed by Kraut et al.⁶⁰. The energy difference between a core level (CL) and homo band maximum (HBM) for individual materials was measured and then the difference

between CLs of CV and [Au₂₅(Cys)₁₈] was measured. Subsequently, the band alignment and energy offset were determined by equation (2) below.

$$\Delta E_{\text{HBM}} = (E_{\text{CL}}^{\text{Au cluster}} - E_{\text{HBM}}^{\text{Au cluster}}) - (E_{\text{CL}}^{\text{CV}} - E_{\text{HBM}}^{\text{CV}}) - \Delta E_{\text{CL}} \quad (2)$$

Where $E_{\text{CL}}^{\text{Au cluster}}$ and $E_{\text{CL}}^{\text{CV}}$ are the energy of [Au₂₅(Cys)₁₈] and crystal violet core levels, respectively and $E_{\text{HBM}}^{\text{Au cluster}}$ and $E_{\text{HBM}}^{\text{CV}}$ are the energy of [Au₂₅(Cys)₁₈] and crystal violet HBMs, respectively. ΔE_{CL} is $E_{\text{CL}}^{\text{Au cluster}} - E_{\text{CL}}^{\text{CV}}$ and ΔE_{HBM} is $E_{\text{HBM}}^{\text{Au cluster}} - E_{\text{HBM}}^{\text{CV}}$. Additionally, to determine the peaks' position of XPS spectra precisely, Shirley background and Gaussian-Lorentzian profiles were used."

2. An explanation about the alternative photoinactivation pathway should be added to the manuscript. Moreover, no indication about the CV concentration in the polymer with CV and in the polymer with CV/Au is presented. However, the Figure 2c of the supplementary information shows that the concentration of CV in the polymer with only CV is much lower than that in the polymer with CV/Au, at least 2 times lower. This can explain, at least, in part, the difference in bacterial inactivation by the two polymers. The information about the CV concentration in both polymers is essential to interpret the results and should be added to the manuscript and discussed

Response

Corresponding to the reviewer's comment, the addition of Au cluster significantly enhanced CV penetration into the polymer. However, we do not think that the two-fold increase in CV concentration in the polymer caused the bactericidal activity of the sample increased from 0 to ~3 log kill. If the enhancement mechanism was mainly due to an increase of CV concentration, the reaction Type-II should be enhanced. Our $^1\text{O}_2$ phosphorescence measurement clearly shows that reaction Type-II decreased when the Au cluster was added. Thus, it is concluded that the photobactericidal enhancement is mainly due to an interaction between Au clusters and CV molecules. For better understanding, additional experiments were performed, and the results showed that upon white light illumination, photogenerated electron in crystal violet flows from crystal violet to the gold nanocluster. As a result, excessive electron accumulation in the cluster causes electron transfer to the environment and promoted redox reactions. The results and discussion of the photoreaction mechanism are added into the manuscript (Page 9 line to 11, Figure 5 and 6, and Supplementary Figure 5)

To determine a change of CV concentration in the polymer before and after additional encapsulation of $[\text{Au}_{25}(\text{Cys})_{18}]$, XRF analysis was employed. Among the materials used only crystal violet contains Cl. Thus, a change in weight percentage (wt%) of Cl was determined using XRF. After additional encapsulation of $[\text{Au}_{25}(\text{Cys})_{18}]$, The increase of Cl wt% was nearly twice compared to CV only polymer. This result was stated in manuscript.

(Page 5 and Supplementary Figure 4).

Information as below was added into manuscript

A swell-encapsulation-shrink process was employed to produce photobactericidal silicone. Fluorescence microscopy, UV-Vis spectroscopy and XPS confirmed that after the treatment, CV molecules and $[\text{Au}_{25}(\text{Cys})_{18}]$ clusters penetrated into the polymer matrix (Supplementary Figure 3). As shown in Figure 2a, after $[\text{Au}_{25}(\text{Cys})_{18}]$ or cysteine treatment, the silicone substrate maintained its colour, while CV treatment changed the colour from white to violet. The colour of the polymer containing CV& $[\text{Au}_{25}(\text{Cys})_{18}]$ was more intense than the sample with CV alone. Figure 2b shows UV-vis absorbance spectra of the control and treated samples at a wavelength of 400 to 800 nm. All of the CV-treated polymers have a main absorbance at 595 nm with a shoulder peak at 541 nm. The absorbance of $[\text{Au}_{25}(\text{Cys})_{18}]$ encapsulated sample was broader and of higher intensity compared to the sample with CV alone. Of materials used for the encapsulation, CV is the only material containing chlorine (Cl). Thus, a change in the weight percentage (wt%) of Cl was determined before and after additional encapsulation of $[\text{Au}_{25}(\text{Cys})_{18}]$ by X-ray fluorescence spectroscopy (XRF). Cl wt% in CV& $[\text{Au}_{25}(\text{Cys})_{18}]$ polymer was 1.8 times higher than the polymer containing CV alone. This was similar with the increase of light absorbance of the material at 595 nm after additional encapsulation of $[\text{Au}_{25}(\text{Cys})_{18}]$ into the CV treated polymer. This showed that addition of $[\text{Au}_{25}(\text{Cys})_{18}]$ enhanced CV impregnation into the polymer. The image analysis of sliced polymers supports the increase of CV uptake by $[\text{Au}_{25}(\text{Cys})_{18}]$ (Supplementary Figure 4). We speculate that this is because negatively charged $[\text{Au}_{25}(\text{Cys})_{18}]$ (zeta potential of $[\text{Au}_{25}(\text{Cys})_{18}]$ solution: -31.8 mV) attracts more CV⁺ in during the encapsulation process.³⁷

”

Supplementary Figure 4 Distribution of crystal violet in CV only and CV&[Au₂₅(Cys)₁₈] polymer. a, Side section image of sliced CV only and CV&[Au₂₅(Cys)₁₈] polymers after 1, 5 and 24 h encapsulation. **b,** Profile of CV distribution inside CV only and CV&[Au₂₅(Cys)₁₈] polymers after 1, 5 and 24 h encapsulation.

The side images of thinly sliced CV only and CV&[Au₂₅(Cys)₁₈] polymers were taken by optical microscope and they were analysed using ImageJ. As shown in Supplementary Figure 4, CV diffusion into the silicone was accelerated after additional encapsulation of [Au₂₅(Cys)₁₈] and after 24 h encapsulation, the polymer containing [Au₂₅(Cys)₁₈] exhibited more intense violet colour indicating that silicone with CV & [Au₂₅(Cys)₁₈] has more CV molecules inside the polymer than the polymer with CV alone.

3. The experiments were done with a Gram-positive bacterium, but it is well known that Gram negative bacteria are more difficult to inactivate by photodynamic therapy than Gram positive ones. No explanation is presented about why the authors choose a Gram-positive bacterium to test the new material.

Answer

I agree with the reviewer's comment. Gram-negative bacteria are more resistant to reactive oxygen species because they have more complex cell wall. There are several Gram-negative bacteria which are associated with hospital-acquired infection including *E. coli* and a representative strain (ATCC 25922) was tested here. The results showed that *E. coli* requires a longer exposure time to white light to obtain a significant bacterial kill compared to *S. aureus*. 2.8 log reduction in the number of *E. coli* bacteria was observed on polymer with CV&[Au₂₅(Cys)₁₈] after 24 h exposure to white light. The experimental result and discussion of *E. coli* were written in the manuscript. (Page 7 and 8, Figure 3 c and d).

Information as below was added into manuscript

“Figures 3c and d show the bactericidal activity of the samples against *E. coli* after 24 h incubation in the dark and in white light. After 24 h incubation in the dark, compared to the control, the reduction in the number of viable bacteria was not statistically significant on the samples with only CV or [Au₂₅(Cys)₁₈] alone or CV&[Au₂₅(Cys)₁₈] ($P > 0.1$), while a significant reduction in the numbers of viable bacteria was observed on the sample containing cysteine only (0.7 log reduction, $P < 0.1$). After 24 h exposure to white light, the polymer samples containing either CV or [Au₂₅(Cys)₁₈] alone did not show any bactericidal activity compared to the control, and the sample with cysteine only did not show any increase in bactericidal activity compared to that in the dark ($P > 0.1$). However, bactericidal activity of the polymer with CV&[Au₂₅(Cys)₁₈] was significantly enhanced after 24 h exposure of white light; compared to the control, showing a 2.8 log reduction in the number of viable bacteria ($P < 0.01$).

E. coli was more resistant than *S. aureus* requiring a longer exposure time of white light to achieve a significant reduction in the number of viable bacteria. We attribute this to differences in the cell wall, where *E. coli* - a Gram-negative bacterium - contains a double membrane structure compared to *S. aureus* - a Gram-positive bacterium - which contains only a single membrane barrier.⁴⁰ The outer membrane of Gram-negative bacterium decreases molecular penetration, and is often responsible for increased resistance to antibacterial agents.^{40,41}”

Figure 3 Bactericidal activity of control and treated silicone samples. **c.** Bactericidal activity of control, [Au₂₅(Cys)₁₈] only, CV only, Cysteine only and CV&[Au₂₅(Cys)₁₈] against *E. coli* in dark. **d.** Bactericidal activity of control, [Au₂₅(Cys)₁₈] only, CV only, Cysteine only and CV&[Au₂₅(Cys)₁₈] of *E. coli* in white light. Bacteria inoculated materials were exposed to white light with an intensity from 200 to 429 lux.

4. There is no indication about the dark period of incubation before the phototreatment. The treatment was done on a solid surface, but the results are expressed by mL. Which was the bacterial decrease by cm² of the polymer?

Answer

We have done bacteria test under dark conditions with identical incubation time to those experiments under light condition (Figure 4a and c and Supplementary Figure 7). The results show limited bactericidal activity samples before light exposure. In our research, 25 uL (one droplet) of bacteria suspension was inoculated on the sample surface, and then a reduction on the number of viable bacteria in the solution was determined. This method is widely used to investigate bactericidal activity against solid samples^{1,2}.

1. Dunhill et al. Journal of Materials Chemistry 2009, 19, 8747–8754

2. Perni et al. Biomaterials 2009, 30, 89–93

We do not spread bacteria on the samples. As shown in the figure, bacterial suspension was an inoculated sample. The suspension forms a hemisphere. Thus, it would be more appropriate that Bacterial distribution is three-dimension rather than two-dimension. In our research, it would not be appropriate to express bacterial by number/cm².

5. The light intensity used in the study is expressed in lux, but to compare with other studies already described in the literature the intensity should be also expressed in mw/cm².

Response

We also expressed intensity in mW/cm² (Page3, 6, 11, and Supplementary Figure 5)

6, How was the bacterial inoculum spread at the surface of the materials? There is no indication in the supplementary information.

Answer

As shown figure below, bacteria suspension was not spread on the sample. 25 uL of bacterial suspension was inoculated and keep on the surface. For better understanding of antimicrobial protocol, an additional figure was added into the Supplementary information.

Supplementary Figure 7 | Procedure of bactericidal test

7. No indication about the number of independent assays done is indicated. The number of replicates per dilution is also not presented.

Response

Each experiment contained 2 technical replicates and the experiment was reproduced three times. This information was added into Material and methods of manuscript (page 15 and 16).

Information as below was added into manuscript

“Each experiment contained 2 technical replicates and the experiment was reproduced three times”

8. The detection limit of the method used to determine the bacterial concentration is too high $< 10^2$ CFU/mL. Is it correct?

Response

We have checked our calculations and the detection limit expressed per ml is correct. This is equivalent to 1 colony in a 200 μ l sample which was the volume plated

9. The authors stated that “The ROS and $^1\text{O}_2$, produce”, but the $^1\text{O}_2$ is a ROS.

Response

As reviewer’s comment, $^1\text{O}_2$ is ROS. However, because ROS identities induced from crystal violet were not totally determined. So, we used “ROS and $^1\text{O}_2$ ” at same time. We revised the terms into “ $^1\text{O}_2$ and other ROS” (Page 7).

10. Throughout the manuscript in many cases a “,” is used before the “and”, in these cases the “,” should be removed.

Response

As reviewer’s comment, we remove “,” before the “and”.

11. I cannot understand the Figure 3a of the supplementary information.

Response

We revised Supplementary Figure 5a for better understanding as below.

12. According to Figure 4b of the supplementary information, around 10% of the used light is UV, but in the manuscript it is indicated that the experiments were done with white light.

Answer

The white lamp that we used is widely used in hospital and other healthcare facilities. White light normally emits a small amount of UV. Compared to light emission in visible range, the UV portion is quite low. This feature is also shown in other white lamps as below.

Light emission spectrum of Osram Orbeos CMW-301
P. A. Haigh and Z. Ghassemlooy, IEEE Photonics Technology Letters · March 2013

Light emission spectrum of GE lighting fluorescent lamp SPX50 HL

Y-axis of Supplementary Figure 5b is normalized by light emission power at 550 nm. It does not mean that UV accounts for 10% of total emission of the white light. As below we revised the Figure 5b to prevent readers from being confused.

Reviewer #2 (Remarks to the Author):

1. Photobactericidal activity of crystal violet encapsulated silicone activated by thiolated gold nanoclusters at low flux levels of white light

The above title manuscript is very preliminary work not worth publishing Nature Communication.

General comments:

What is the novelty of the work as this kind of work has already been published.

1. Naik AJ, Ismail S, Kay C, Wilson M, Parkin IP. Antimicrobial activity of polyurethane embedded with methylene blue, toluidene blue and gold nanoparticles against *Staphylococcus aureus*; illuminated with white light. *Materials Chemistry and Physics*. 2011 Sep 15;129(1-2):446-50.

2. Perni S, Piccirillo C, Pratten J, Prokopovich P, Chrzanowski W, Parkin IP, Wilson M. The antimicrobial properties of light-activated polymers containing methylene blue and gold nanoparticles. *Biomaterials*. 2009 Jan 1;30(1):89-93. What is the rationale behind this work?, Not clear.

Response

Contrary to water splitting, photoelectrical cell, and water cleaning, light source should be carefully considered for use for photocatalytic disinfection in hospital or other indoor facilities. Firstly, the light source must not make hospital staff and patients uncomfortable or produce adverse effects like an Ultraviolet light source. Secondly, white light sources ranging from 200 to 90000 lux are widely used in hospitals and healthcare facilities. Thus, photocatalytic disinfection should be shown under various light conditions indicating that the sample should maintain photobactericidal activity under low white light levels.

As the referee's comment indicates, we have published several papers in terms of light activated polymer with Au nanoparticle. However, all of the papers showed enhanced photobactericidal activity under laser radiation or intense white light (>1000 lux), and the polymer containing Au nanoparticles and did not represent or reinforced photobactericidal activity under low flux levels of white light. Notably this work reports the first use of a discrete chemical entity a gold cluster of extremely well-defined composition and significantly smaller than any gold nanoparticle we have used previously. Gold (Au) materials can be classified into three different levels containing bulk, nanoparticle, and atomic cluster. Bulk Au are electrical conductors and good optical reflectors. Gold nanoparticles appears intense in colour in solution because of surface plasmon resonance. Metal nanocluster consisting of a small number of atoms are known as bridging link between atoms and nanoparticles. Its size is typically less than 2 nm or containing less than 40 Au atoms, and it has been known that the cluster does not have plasmonic behaviour in contrast to nanoparticles. Electronic band structure of Au nanocluster is difference from Au nanoparticles; the band structure of Au nanoparticles is continuous while Au nanocluster has a discontinuous band structure indicating that it has discrete energy levels.

Because Au cluster used in this study has different physico-chemical property from Au nanoparticles, it had been expected that Au cluster might produce different results from that of Au nanoparticles. As a result, the cluster added polymer represented potent photobactericidal activity under low flux level of white light. To the best of our knowledge, this is unprecedented and a big step forward in photocatalytic study for real world application. Additionally, a novelty and justification of our research were discussed in our manuscript (page 11)

Information as below was added into manuscript

“For the widespread use of photobactericidal coatings in hospitals and other indoor facilities, the light source should be carefully considered. Firstly, the light source should not produce an adverse effect on hospital staff and patients. Secondly, because light sources typically range from 300 (corridors, rooms) to 90,000 lux (operating theatre) in hospitals and healthcare facilities (Table 1), potent photobactericidal activity should be present in various lighting conditions. Titanium dioxide (TiO₂) and zinc oxide (ZnO) nanoparticles are the most extensively studied photocatalysts, and they exhibit a broad photobactericidal spectrum.⁵¹⁻⁵³ However, their bactericidal activity is negligible under white light sources, which are commonly used in healthcare facilities, because TiO₂ and ZnO nanoparticles are UV-active photocatalysts, and the portion of UV light present in indoor lighting is typically extremely low.⁵¹ To produce enhanced indoor photocatalysts, TiO₂ based nanocomposites using Ag NPs and Au NPs, carbon nanotube (CNT), and graphene oxide (GO) *etc* have been studied.⁵²⁻⁵⁴ Although some of the composites exhibited bactericidal activity under a white light source, such studies typically employed an intense light source of >1300 lux (0.2 mW/cm²) indicating lack of feasibility for use in healthcare facilities (Table 1)²⁵²⁻⁵⁷. In recent years, it was reported that TBO, CV, and MB treated polymer surfaces exhibit photobactericidal activity and the addition of 2 nm Au NPs or 3 nm ZnO NPs within the treated surface results in enhanced photobactericidal activity.^{24,26,29,58} However, intense white light of >1000 lux (>0.15 mW/cm²) or a ~600 nm laser source had to be employed to achieve potent bacterial kill, indicating that such polymers can only be used in healthcare environments with extremely bright lighting, such as operating theatres and A&E examination units (Table 1).^{24,26,29,58,59} Our study showed that the addition of [Au₂₅(Cys)₁₈] into a CV impregnated polymer significantly enhances its photobactericidal activity, showing potency under low intensity white light ranging from 200 to 429 lux (0.03 to 0.06 mW/cm²), which is >3 times lower light levels than previous studies.^{24,26,29,52-54,58,59}”

2. What is the mechanism of killing? The author stated on line number 67 that “gold cluster could promote a greater synergistic effect “ if author was checking synergistic effect then they should go for checker board assay.

Response

To address the comment of reviewer 2, additional experiments containing photocurrent measurement, XPS, Time resolved PL and steady state PL spectroscopy were conducted. The results showed that upon white light illumination, photogenerated electrons in crystal violet flows from crystal violet to the gold nanocluster. As a result, excessive electron accumulation in the cluster cause electron transfer to the environment and promoted redox reactions. This indicate that the addition of Au cluster produces alternative pathway of electron transfer and enhance redox reaction. As a result, photochemical reaction Type-I was reduced. The results and discussion of the photoreaction mechanism were added into manuscript (Page 9 to 11 and 16 to 17, Figure 5 and 6, and Supplementary Figure 6).

Information as below was added into manuscript

[Au₂₅(Cys)₁₈] to CV reduces the Type-II pathway

“To understand the photoreaction mechanism of the polymer sample containing CV&[Au₂₅(Cys)₁₈], photocurrent measurements, steady state and time-resolved photoluminescence (PL) spectroscopies were employed. Transient photocurrent responses of CV only and CV&[Au₂₅(Cys)₁₈] treated polymers were measured under several on-off cycles of white light irradiation. As shown in Figure 5a, a greater

rise in photocurrent of the polymers containing CV&[Au₂₅(Cys)₁₈] compared to CV alone was observed, indicating a higher separation efficiency of electron-hole pairs. Figure 5b shows PL spectra of the samples with CV only and CV&[Au₂₅(Cys)₁₈] from 600 to 800 nm. The intensity of the PL spectrum is a direct measurement on recombination rate of electron-hole pair. The higher the peak intensity of the spectrum, the easier the recombination of the electron-hole pair. A PL peak of the polymer sample with CV only was observed at ~675 nm, and the PL peak of the sample with CV&[Au₂₅(Cys)₁₈] was lower than that of CV only indicating that the recombination rate of the photogenerated electron-hole pairs on CV&[Au₂₅(Cys)₁₈] is slower than that of CV only. This indicates that additional encapsulation of [Au₂₅(Cys)₁₈] into CV treated polymer improves the photoelectrochemical properties of the sample. Figure 5c shows time-resolved PL decay of samples containing CV only or CV&[Au₂₅(Cys)₁₈]. The PL decay was measured at a wavelength of 650 nm upon excitation by a laser source with a wavelength of 574 nm. Compared to the CV only polymer, the PL life time of the polymer sample with CV&[Au₂₅(Cys)₁₈] was shorter. This was attributed to electron transfer from CV to [Au₂₅(Cys)₁₈].^{47,48} It is suggested that [Au₂₅(Cys)₁₈] on the CV treated surface acts as electron acceptor, resulting in better electron-hole pair separation, a reduction in recombination, and enhanced photobactericidal properties.^{47,48}

In order to determine the band offset of the crystal violet and [Au₂₅(Cys)₁₈], XPS and UV-Vis spectroscopy were used, with the results shown in Figure 6. For XPS analysis, the adventitious C 1s peak at 284.8 eV was used as a reference for charge correction. Band gap energies (E_G) of crystal violet and [Au₂₅(Cys)₁₈] were 1.95 and 1.25 eV, respectively (Supplementary Figures 6a and d) and homo band maximum energies (E^{HBM}) of crystal violet and [Au₂₅(Cys)₁₈] were 1.6 and 1.07 eV, respectively (Supplementary Figures 6b and e). A difference of binding energy between N 1s core level (CL) and homo band was investigated in CV only and [Au₂₅(Cys)₁₈] only samples (Supplementary Figures 6c and f). The same functions were used to peak fit N 1s peaks in CV&[Au₂₅(Cys)₁₈] to determine the binding energy (Figure 6a). As shown in Figure 6b, the difference of binding energy between N 1s CL and E^{HBM} was 397.78 eV for CV and 397.81 eV for [Au₂₅(Cys)₁₈]. The E^{HBM} of [Au₂₅(Cys)₁₈] was 0.53 eV higher than that of CV while the lumo band maximum energy (E^{LBM}) of crystal violet was 0.17 eV higher than that of [Au₂₅(Cys)₁₈], indicating the formation of a straddling (type I) band alignment at interface between CV and [Au₂₅(Cys)₁₈].^{49,50} This alignment suggests, that upon white light illumination, photogenerated electrons can flow from CV to [Au₂₅(Cys)₁₈], resulting in electron accumulation in [Au₂₅(Cys)₁₈]; in agreement with time-resolved PL measurements (Figure 5c)."

Figure 5 Photoreaction characterisation of CV only and CV&[Au₂₅(Cys)₁₈]. **a**, Transient photocurrent responses for CV only and CV&[Au₂₅(Cys)₁₈] under white light. **b**, photoluminescence (PL) spectra of CV only and CV&[Au₂₅(Cys)₁₈] in wavelength of 600 to 850 nm ($\lambda_{Ex} = 574$ nm). **c**, Time-resolved photoluminescence (PL) decay of CV only and CV&[Au₂₅(Cys)₁₈] ($\lambda_{Ex} = 574$ nm, $\lambda_{Em} = 650$ nm).

Figure 6 band alignment of crystal violet and [Au₂₅(Cys)₁₈] from XPS. **a**, N 1s spectra taken from CV only, [Au₂₅(Cys)₁₈] only and CV&[Au₂₅(Cys)₁₈]. Experiment data shown as blue line are fitted with the peak shapes derived from phase-pure crystal violet (red line) and [Au₂₅(Cys)₁₈] (black line). **b**, XPS band alignment between crystal violet and [Au₂₅(Cys)₁₈]. ΔE_{CL} is the core level offset between the N 1s core levels.

Supplementary Figure 6 | band gap, homo band and N 1s spectra from phase-pure crystal violet and [Au₂₅(Cys)₁₈]. a, band gap, b, homo band and c, N 1s spectra for phase-pure crystal violet. d, band gap, b, homo band and c, N 1s spectra for phase pure [Au₂₅(Cys)₁₈].

Materials and method

Photoelectrochemical measurement and photoluminescence spectroscopy. Photocurrent measurements were performed using Metrohm Autolab (PGSTAT302N, Utrecht, Netherlands) with a three-electrode system with an external source of white light. Ag/AgCl reference electrode, and platinum sheet as a counter electrode. The photobactericidal polymer (CV only and CV&[Au₂₅(Cys)₁₈]) served as the working electrode. The electrolyte used in the system was 0.1 M Na₂SO₄.

Photoluminescence (PL) spectra of samples were measured in a wavelength of 600 – 800 nm using steady state PL spectrometer (FluoroMax, Horiba Scientific, Kyoto, Japan). The excitation wavelength was ~574 nm.

Time-resolved photoluminescence spectroscopy. Time-resolved photoluminescence spectroscopy, for timescales up to 100 ns (~24.4 ps resolution), was measured using a time-correlated single photo counting (TCSPC) apparatus (DeltaFlex, Horiba Scientific). Pulsed 574 nm excitation (1 Mz repetition rate, <1.6 ns pulse width) was generated by a laser diode (NanoLED-570), and the fluorescence was detected at wavelengths at 650 nm (Picosecond Photon Detection Module, PPD-900, Horiba Scientific, Kyoto, Japan).

Zeta potential measurement

Zeta potential of the [Au₂₅(Cys)₁₈] in the mixture of water and acetone were measured by Delsa zeta potential analyzer (DelsaMax-Pro, Beckman Coulter) with the available flow cell system in batch mode at 22 °C.

Measurement of band offset

XPS was employed to determine the band offset of crystal violet and [Au₂₅(Cys)₁₈] using the method proposed by Kraut et al.⁶⁰. The energy difference between a core level (CL) and homo band maximum (HBM) for individual materials was measured and then the difference between CLs of CV and [Au₂₅(Cys)₁₈] was measured. Subsequently, the band alignment and energy offset were determined by equation (2) below.

$$\Delta E_{\text{HBM}} = (E_{\text{CL}}^{\text{Au cluster}} - E_{\text{HBM}}^{\text{Au cluster}}) - (E_{\text{CL}}^{\text{CV}} - E_{\text{HBM}}^{\text{CV}}) - \Delta E_{\text{CL}} \quad (2)$$

Where $E_{\text{CL}}^{\text{Au cluster}}$ and $E_{\text{CL}}^{\text{CV}}$ are the energy of [Au₂₅(Cys)₁₈] and crystal violet core levels, respectively and $E_{\text{HBM}}^{\text{Au cluster}}$ and $E_{\text{HBM}}^{\text{CV}}$ are the energy of [Au₂₅(Cys)₁₈] and crystal violet HBMs, respectively. ΔE_{CL} is $E_{\text{CL}}^{\text{Au cluster}} - E_{\text{CL}}^{\text{CV}}$ and ΔE_{HBM} is

$E_{\text{HBM}}^{\text{Au cluster}} - E_{\text{HBM}}^{\text{CV}}$. Additionally, to determine the peaks' position of XPS spectra precisely, Shirley background and Gaussian-Lorentzian profiles were used."

3. Photobactericidal mechanism is due to which compound? There is no killing after light activation in [Au₂₅(Cys₁₈)], CV and Cysteine then what happen to CV&[Au₂₅(Cys₁₈)]? What is the mechanism behind the Photobactericidal activity of combined CV&[Au₂₅(Cys₁₈)]? Overall this manuscript lack novelty as well as they fail to justify their own work.

Response

Photobactericidal enhancement is mainly due to the gold nanocluster. As mentioned above, the detail of mechanism was added into manuscript (Page 9 to 11 and 16 to 17, Figure 5 and 6, and Supplementary Figure 6)

As a response to the first comment, we explained the novelty of our research and justification.

Additionally, a novelty and justification of our research were discussed in our manuscript.

Information as below was added into manuscript

“For the widespread use of photobactericidal coatings in hospitals and other indoor facilities, the light source should be carefully considered. Firstly, the light source should not produce an adverse effect on hospital staff and patients. Secondly, because light sources typically range from 300 (corridors, rooms) to 90,000 lux (operating theatre) in hospitals and healthcare facilities (Table 1), potent photobactericidal activity should be present in various lighting conditions. Titanium dioxide (TiO₂) and zinc oxide (ZnO) nanoparticles are the most extensively studied photocatalysts, and they exhibit a broad photobactericidal spectrum.⁵¹⁻⁵³ However, their bactericidal activity is negligible under white light sources, which are commonly used in healthcare facilities, because TiO₂ and ZnO nanoparticles are UV-active photocatalysts, and the portion of UV light present in indoor lighting is typically extremely low.⁵¹ To produce enhanced indoor photocatalysts, TiO₂ based nanocomposites using Ag NPs and Au NPs, carbon nanotube (CNT), and graphene oxide (GO) *etc* have been studied.⁵²⁻⁵⁴ Although some of the composites exhibited bactericidal activity under a white light source, such studies typically employed an intense light source of >1300 lux (0.2 mW/cm²) indicating lack of feasibility for use in healthcare facilities (Table 1)²⁵²⁻⁵⁷. In recent years, it was reported that TBO, CV, and MB treated polymer surfaces exhibit photobactericidal activity and the addition of 2 nm Au NPs or 3 nm ZnO NPs within the treated surface results in enhanced photobactericidal activity.^{24,26,29,58} However, intense white light of >1000 lux (>0.15 mW/cm²) or a ~600 nm laser source had to be employed to achieve potent bacterial kill, indicating that such polymers can only be used in healthcare environments with extremely bright lighting, such as operating theatres and A&E examination units (Table 1).^{24,26,29,58,59} Our study showed that the addition of [Au₂₅(Cys)₁₈] into a CV impregnated polymer significantly enhances its photobactericidal activity, showing potency under low intensity white light ranging from 200 to 429 lux (0.03 to 0.06 mW/cm²), which is >3 times lower light levels than previous studies.^{24,26,29,52-54,58,59}”

Reviewers' Comments:

Reviewer #1:

Remarks to the Author:

The authors have taken all of my comments into account, and carefully revised the manuscript. I think that the manuscript is much stronger now.

Reviewer #2:

Remarks to the Author:

What is the novelty of the work as this kind of work has already been published.

1. Naik AJ, Ismail S, Kay C, Wilson M, Parkin IP. Antimicrobial activity of polyurethane embedded with methylene blue, toluidene blue and gold nanoparticles against *Staphylococcus aureus*; illuminated with white light. *Materials Chemistry and Physics*. 2011 Sep 15;129(1-2):446-50.

2. Perni S, Piccirillo C, Pratten J, Prokopovich P, Chrzanowski W, Parkin IP, Wilson M. The antimicrobial properties of light-activated polymers containing methylene blue and gold nanoparticles. *Biomaterials*. 2009 Jan 1;30(1):89-93.

What is the rationale behind this work?

What is the mechanism of killing? The author stated on line number 67 that "gold cluster could promote a greater synergistic effect " if author was checking synergistic effect then they should go for checker board assay.

Photobactericidal mechanism is due to which compound? There is no killing after light activation in Au₂₅(Cys)₁₀, CV and Cysteine then what happen to CV&Au₂₅(Cys)₁₀? What is the mechanism behind the Photobactericidal activity of combined CV&Au₂₅(Cys)₁₀?

Overall this manuscript lack novelty as well as they fails to justify their own work.

Reviewers' comments:

Reviewer #1 (Remarks to the Author):

The authors have taken all of my comments into account, and carefully revised the manuscript. I think that the manuscript is much stronger now.

Answer

We really appreciate the comment of reviewer 1. We are confident that this research has sufficient novelty to be published in Nature communication.

Reviewer #2 (Remarks to the Author):

What is the novelty of the work as this kind of work has already been published.

1. Naik AJ, Ismail S, Kay C, Wilson M, Parkin IP. Antimicrobial activity of polyurethane embedded with methylene blue, toluidene blue and gold nanoparticles against *Staphylococcus aureus*; illuminated with white light. *Materials Chemistry and Physics*. 2011 Sep 15;129(1-2):446-50.

2. Perni S, Piccirillo C, Pratten J, Prokopovich P, Chrzanowski W, Parkin IP, Wilson M. The antimicrobial properties of light-activated polymers containing methylene blue and gold nanoparticles. *Biomaterials*. 2009 Jan 1;30(1):89-93.

Answer

As the referee's comment indicates, we have published several papers in terms of light activated polymer with Au nanoparticle. However, all of the papers showed enhanced photobactericidal activity under laser radiation or intense white light (>1000 lux), and the polymer containing Au nanoparticles (typically ca 1000- 100,000 atoms in size) did not represent or show photobactericidal activity under low flux levels of white light. **Notably this work reports the very first use of a discrete chemical entity a gold cluster** of extremely well-defined composition (25 gold atoms) and significantly smaller than any gold nanoparticle we have used previously. Note previous gold nanoparticles had a range of sizes (2nm to 20nm) and were not single chemical entities- ie they were not monodisperse. Gold (Au) materials can be classified into three different levels containing bulk, nanoparticle, and atomic cluster. Bulk Au are electrical conductors and good optical reflectors. Gold nanoparticles appears intense in colour in solution because of surface plasmon resonance. Metal nanocluster consisting of a small number of atoms are known as bridging link between atoms and nanoparticles. Its size is typically less than 2 nm or containing less than 40 Au atoms, and it has been shown that the cluster does not have plasmonic behaviour in contrast to nanoparticles. Electronic band structure of Au nanocluster is

different from Au nanoparticles; the band structure of Au nanoparticles is continuous while Au nanocluster has a discontinuous band structure indicating that it has discrete energy levels.

Because Au cluster used in this study has different physico-chemical property from Au nanoparticles, it had been expected that Au cluster might produce different results from that of Au nanoparticles. As a result, the cluster added polymer represented potent photobactericidal activity under **low** flux level of white light. To the best of our knowledge, this is unprecedented and a big step forward in photocatalytic study for real world application. The novelty of our work is discussed on pages 10-11 of the manuscript.

2. What is the rationale behind this work? What is the mechanism of killing? The author stated on line number 67 that “gold cluster could promote a greater synergistic effect “ if author was checking synergistic effect then they should go for checker board assay.

Answer

We have already reported extensively on this in the revised manuscript. The reviewer seems not to have seen or understood these changes. We report extensively and did a wide range of new tests to comment on the mechanism. We have effectively done and reported a checker board assay by measuring the activities of kill of the various control samples and for two bacteria types at different concentrations.

To understand photobactericidal mechanism of CV&[Au₂₅(Cys)₁₈], a variety of experimental assay were used. Firstly, to determine ROS responsible for the bactericidal effect observed in our CV&[Au₂₅(Cys)₁₈], ROS scavenger/quencher assays containing superoxide dismutase, mannitol, L-histidine and catalase and ¹O₂ phosphorescence measurements were carried out. Secondly, experiments containing photocurrent measurement, XPS, time resolved PL and steady state PL spectroscopy were conducted in order to understand the interaction of the gold cluster and crystal violet. Figure 4, 5 and 6 shows the results of the mechanism assay (Page 30 to 33)

The results showed that upon white light illumination, photogenerated electrons in crystal violet flows from crystal violet to the gold nanocluster. As a result, excessive electron accumulation in the cluster cause electron transfer to the environment and promoted redox reactions. This indicates that the addition of Au cluster produces alternative pathway of electron transfer and enhance redox reaction and redox reaction promoted hydrogen peroxide which is toxic to bacteria. As a result, photochemical reaction Type-II was reduced. The results and discussion of the photoreaction mechanism were added into manuscript (Page 8 to 10).

3. Photobactericidal mechanism is due to which compound? There is no killing after light activation in Au₂₅(Cys)₁₀, CV and Cysteine then what happen to CV&Au₂₅(Cys)₁₀? What is the mechanism behind the Photobactericidal activity of combined CV&Au₂₅(Cys)₁₀?

Response

Photobactericidal enhancement is mainly due to [Au₂₅(Cys)₁₈]. As mentioned above, the cluster produces alternative pathway of electron transfer resulting in enhanced redox reaction. The enhanced redox reaction promotes hydrogen peroxide which are toxic reactive oxygen species to bacteria (Page 8 to 10).

4. Overall this manuscript lack novelty as well as they fails to justify their own work.

Response

The editors of nature Comms note that they are satisfied with the novelty of the work and they did not ask the reviewer to comment on this in their reply.

The key novelty is that this is the first time ever that a metal cluster (a distinct chemical entity) was shown to enhance the ability of a photoactive dye to kill bacteria.

The mechanism of the process was determined and it was shown that the cluster encourages a hydrogen peroxide formation pathway.

This work shows that bacteria can be killed under real world conditions – with light levels that are actually used in wards and corridors in hospitals (ca 300 lux) rather than the high flux levels used in previous experiments (1000- 8000 lux).

For the widespread use of photobactericidal coatings in hospitals and other indoor facilities, the light source should be carefully considered. Firstly, the light source should not produce an adverse effect on hospital staff and patients. Secondly, because light sources typically range from 300 (corridors, rooms) to 90,000 lux (operating theatre) in hospitals and healthcare facilities (Table 1), potent photobactericidal activity should be present in various lighting conditions. Titanium dioxide (TiO₂) and zinc oxide (ZnO) nanoparticles are the most extensively studied photocatalysts, and they exhibit a broad photobactericidal spectrum. However, their bactericidal activity is negligible under white light sources, which are commonly used in healthcare facilities, because TiO₂ and ZnO nanoparticles are UV-active photocatalysts, and the portion of UV light present in indoor lighting is typically extremely low. To produce enhanced indoor photocatalysts, TiO₂ based nanocomposites using Ag NPs and Au NPs, carbon nanotube (CNT), and graphene oxide (GO) *etc* have been studied. Although some of the

composites exhibited bactericidal activity under a white light source, such studies typically employed an intense light source of >1300 lux (0.2 mW cm^{-2}) indicating lack of feasibility for use in healthcare facilities.

In recent years, it was reported that TBO, CV, and MB treated polymer surfaces exhibit photobactericidal activity and the addition of 2 nm Au NPs or 3 nm ZnO NPs within the treated surface results in enhanced photobactericidal activity. However, intense white light of >1000 lux ($>0.15 \text{ mW cm}^{-2}$) or a ~ 600 nm laser source had to be employed to achieve potent bacterial kill, indicating that such polymers can only be used in healthcare environments with extremely bright lighting, such as operating theatres and A&E examination units (Table 1).

Our study showed that the addition of $[\text{Au}_{25}(\text{Cys})_{18}]$ into a CV impregnated polymer significantly enhances its photobactericidal activity, showing potency under low intensity white light ranging from 200 to 429 lux (0.03 to 0.06 mW cm^{-2}), which is >3 times lower light levels than previous studies.

Discussion above was written in result and discussion of manuscript (Page 10 to 11).

Reviewers' Comments:

Reviewer #1:
None

Editorial note; As reviewer #2 was not able to provide a review for this round reviewer #1 was asked to comment on the authors responses to reviewer #2

In comments to the editor reviewer #1 found the authors have responded appropriately to reviewer #2 and there were no more issues to address before publication.